# Inverse Optimal Control Adapted to the Noise Characteristics of the Human Sensorimotor System

**Matthias Schultheis**∗
Centre for Cognitive Science
Technical University of Darmstadt
Darmstadt, Germany
matthias.schultheis@tu-darmstadt.de

**Dominik Straub**∗
Centre for Cognitive Science
Technical University of Darmstadt
Darmstadt, Germany
dominik.straub@tu-darmstadt.de

**Constantin A. Rothkopf**
Centre for Cognitive Science
Technical University of Darmstadt
Darmstadt, Germany
constantin.rothkopf@tu-darmstadt.de

## Abstract

Computational level explanations based on optimal feedback control with signal-dependent noise have been able to account for a vast array of phenomena in human sensorimotor behavior. However, commonly a cost function needs to be assumed for a task and the optimality of human behavior is evaluated by comparing observed and predicted trajectories. Here, we introduce inverse optimal control with signal-dependent noise, which allows inferring the cost function from observed behavior. To do so, we formalize the problem as a partially observable Markov decision process and distinguish between the agent's and the experimenter's inference problems. Specifically, we derive a probabilistic formulation of the evolution of states and belief states and an approximation to the propagation equation in the linear-quadratic Gaussian problem with signal-dependent noise. We extend the model to the case of partial observability of state variables from the point of view of the experimenter. We show the feasibility of the approach through validation on synthetic data and application to experimental data. Our approach enables recovering the costs and benefits implicit in human sequential sensorimotor behavior, thereby reconciling normative and descriptive approaches in a computational framework.

## 1 Introduction

Computational level theories of behavior strive to answer the questions, why a system behaves the way it does and what the goal of the system's computations is. Such goals can be formalized based on the reward hypothesis. In the words of Richard Sutton, the reward hypothesis assumes, that "goals and purposes can be well thought of as maximization of the expected value of the cumulative sum of a received scalar signal" [1]. Thus, to understand human sensorimotor behavior, it is essential to characterize its goals and purposes quantitatively in terms of costs and benefits. But particularly in every-day tasks, the cost and benefits underlying behavior are unknown.

Stochastic optimal control allows formulating behavioral goals in terms of a cost function for tasks involving sequential actions under action variability, uncertainty in the internal model, and delayed rewards. The solution to the optimization problem entailed in the cost function is a sequence of

---

∗Both authors contributed equally to this paper.

35th Conference on Neural Information Processing Systems (NeurIPS 2021)

actions, which is then compared to human movements. While early models optimized costs related to deterministic kinematics of a movement to a target [2], task goals were subsequently formalized as costs on the variance of stochastic movements' endpoint distances to a goal target [3]. Importantly, although [3] considered open-loop control, it revealed the importance of modeling the specific variability of human movements, which increases linearly with the magnitude of the control signal [4]. As the neuronal control signal increases, so does its variability, leading to optimal movements trading off between achieving task goals and reducing the expected impact of movement variability.

Including sensory feedback in stochastic optimal control leads to a computationally much more intricate problem, which can be formulated as a partially observable Markov decision process (POMDP) and is intractable in general. One of the few tractable cases is the linear-quadratic Gaussian (LQG) setting, where dynamics are linear, costs are quadratic, and variability is additive and Gaussian, leading to sensory inference of the state and control to be decoupled [5]. However, not only is human movement variability signal dependent, but additionally the uncertainty of sensory signals increases linearly with the magnitude of the stimulus, a phenomenon known as Weber's law [6].

Todorov [7] extended the LQG case by introducing stochastic optimal feedback control with signal-dependent noise, which allows the specification of noise models in line with what is known about the human sensorimotor system. This model [8] has been able to explain a broad range of phenomena in sensorimotor control [9, 10], including linear movement trajectories, smooth velocity profiles, speed-accuracy tradeoffs, and corrections of errors only if they influence attaining the behavioral goal. Particularly incorporating signal-dependent noise has been crucial in explaining experimental data, ranging from how corrections of movements during action execution depend on feedback and task goals [11], that movements consider sensory uncertainty and temporal delays in real-time [12], that movement plans in novel environments are reoptimized based on the learning of internal models to minimize implicit motor costs and maximize rewards [13], and many others [14, 15, 16, 17].

Explaining human behavior in these studies usually starts by hypothesizing the cost function describing a task, obtaining the optimal feedback controller, and comparing simulated trajectories to those observed experimentally. This line of inquiry, therefore, utilizes similarity of trajectories to quantify the degree of optimality in human behaviors. In some cases [14], trajectories are simulated from the model to check for robustness with respect to changes in the model parameters. If our goal is to use optimal control models to infer such quantities, which often cannot be measured independently, from behavior, it would instead be desirable to invert the problem and find those parameter settings which are consistent with the observed trajectories. While such inverse methods have been developed both in the field of reinforcement learning to infer the rewards being optimized by an agent [18, 19, 20] as well as in optimal control for the LQG case [21, 22, 23], this is currently not possible for the noise characteristics of the human sensorimotor system.

Here, we introduce a probabilistic formulation of inverse optimal feedback control under signal-dependent noise in the tradition of rational analysis [24, 25]. Our starting point is the forward problem introduced in [8]. We formulate the inference problem faced by an agent as a POMDP and distinguish it from the inference problem of an experimenter observing the agent. We proceed by deriving the likelihood of a sequence of observed states and provide an approximation to the non-Gaussian uncertainty due to the signal-dependent noise. First, this allows recovering the cost function underlying the agent's behavior from observed behavioral data. Second, we extend the inference of the cost function to the case in which the state variables are only partially observable to the experimenter, e.g., when only measuring the position of the agent's movement. Third, we show through simulated and experimental data that the cost functions can indeed be recovered. Fourth, the probabilistic formulation allows recovering the agent's belief during the experiment as well as the experimenter's uncertainty about the inferred belief.

## Related work

Inferring the cost functions underlying an agent's behavior has long been of interest in different scientific fields ranging from economics [26] and psychology [27] to neuroscience [28] and artificial intelligence, particularly reinforcement learning [18, 19, 29, 20]. Inverse Reinforcement Learning (IRL) specifically addresses the question of inferring the cost function being optimized [18, 19, 29] or approximately optimized [30] by an agent, with more recent approaches employing deep neural networks [31, 32]. Some work has particularly addressed sensorimotor behavior [33, 34, 35, 36] and extensions to the partially observable setting have been developed [37].

More relevant to the present study are computational frameworks that invert Bayesian models of perception and decision making to infer beliefs and costs of the agent. While this literature is extensive, exemplary studies for trial-based actions include cognitive tomography [38] and inference of the cost function in sensorimotor learning [39]. Extensions to sequential tasks have also been proposed, particularly considering active perception [40, 41], real-world behavior [42, 43], and general formulations [44, 45]. Very similarly, work on Bayesian theory of mind uses highly related computational models, which also allow for the subjective beliefs of the agent to be different from the observer's [46, 47].

In the context of inverse optimal control, related work has considered different problem settings, e.g., deterministic MDPs without additive noise and full observability [48]. More specifically in the LQR and LQG domain, [21] is concerned with finding the cost matrices and noise covariances given a known system with controller and Kalman gain. Similarly, [49] infers the above-mentioned cost matrices in the LQG setting based on observations but additionally takes constraints into account. The extension by [23] infers the terminal cost and the state cost function together with an exponential discount factor in the LQR setting. A different line of work has been concerned with estimating the dynamics, state sequence, and delay of internal LQG models from neural population activity [22, 50] under the assumption that an agent might not know the dynamics, e.g., of a brain-computer interface. Other work [51] is concerned with learning a control policy from states, observations, and controls.

Our approach is different from previous research in two important ways. First, while most other approaches take different quantities such as the filter or controller gains or the agent's observations or controls as given, we consider the setting that is typical in a behavioral experiment: The filter and controller as well as the agent's observations are internal to the experimental subject and cannot be observed. Instead, we assume that only trajectories $\boldsymbol{x}_{1:T}$ are observed. Second, other approaches to inverse optimal control in the LQG setting do not involve signal-dependent noise, which we address in this paper.

## 2    Background: LQG with sensorimorotor noise characterisitics

We model a human subject as an agent in a partially-observable environment as introduced by Todorov [7] and depicted in Fig. 1 A. For this we consider a discrete-time linear dynamical system with state $\boldsymbol{x}_t \in \mathbb{R}^m$ and control $\boldsymbol{u}_t \in \mathbb{R}^p$ with both control-independent and control-dependent noises

$$\boldsymbol{x}_{t+1} = A\boldsymbol{x}_t + B\boldsymbol{u}_t + V\boldsymbol{\xi}_t + \sum_{i=1}^{c} \varepsilon_t^i C_i \boldsymbol{u}_t. \tag{1}$$

The noise terms $\boldsymbol{\xi}_t \in \mathbb{R}^m$ and $\epsilon_t^i \in \mathbb{R}$ are standard Gaussian random vectors and variables, respectively, resulting in control-independent noise with covariance $VV^T$ and control-dependent noise having covariance $\sum_i C_i \boldsymbol{u}_t \boldsymbol{u}_t^T C_i^T$. The agent receives an observation $\boldsymbol{y}_t \in \mathbb{R}^k$ from the observation model

$$\boldsymbol{y}_t = H\boldsymbol{x}_t + W\boldsymbol{\omega}_t + \sum_{i=1}^{d} \epsilon_t^i D_i \boldsymbol{x}_t. \tag{2}$$

The noise terms $\boldsymbol{\omega}_t \in \mathbb{R}^k$ and $\varepsilon_t \in \mathbb{R}$ are again standard Gaussian, so that the covariance of the state-independent observation noise is $WW^T$, while for the state-dependent observation noise it is $\sum_i D_i \boldsymbol{x}_t \boldsymbol{x}_t^T D_i^T$. All matrices of the linear dynamical system can in principle be time-varying, but we leave out the time indices for notational simplicity. The objective of the agent is to choose $\boldsymbol{u}_t$ to minimize a quadratic cost function,

$$J(\boldsymbol{u}_{1:T}) = \mathbb{E}_{\boldsymbol{x}_{1:T}} \left[ \sum_{t=1}^{T} \boldsymbol{x}_t^T Q_t \boldsymbol{x}_t + \boldsymbol{u}_t^T R_t \boldsymbol{u}_t \right]. \tag{3}$$

While the original LQG problem without control- and state-dependent noises can be solved exactly by determining an optimal linear filter and controller independently [5], this separation principle is no longer applicable in the case considered here. Todorov [7] introduced an approximate solution method in which the optimal filters $K_t$ and controllers $L_t$ are iteratively determined in an alternating fashion, leaving the respective other one constant. The resulting optimal filter which minimizes the expected cost, is of the form

$$\tilde{\boldsymbol{x}}_{t+1} = A\tilde{\boldsymbol{x}}_t + B\boldsymbol{u}_t + K_t(\boldsymbol{y}_t - H\tilde{\boldsymbol{x}}_t) + E\boldsymbol{\eta}_t, \tag{4}$$

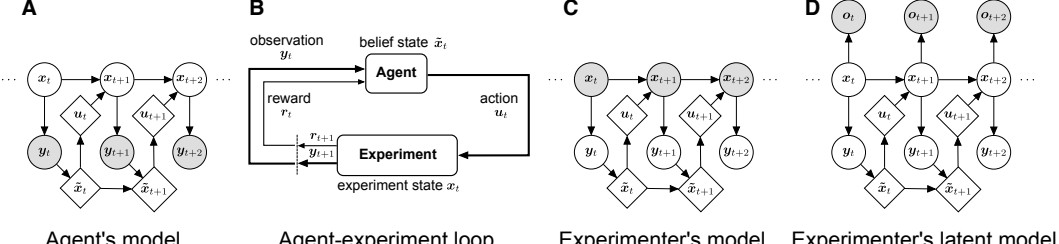

Figure 1: The agent-experimenter loop and its formalization as POMDP together with the inference problems from the agent's point of view and from the experimenter's point of view.

where $\boldsymbol{\eta}_t$ is a standard Gaussian random vector and represents internal estimation noise. The optimal linear control law can be formulated as

$$\boldsymbol{u}_t = -L_t \tilde{\boldsymbol{x}}_t. \tag{5}$$

The equations for determining the matrices $L_t$ and $K_t$ are given in Appendix B. For a detailed derivation the reader is referred to [7].

## 3 Inverse optimal control

In this paper, we consider the inverse problem, i.e., we observe an agent who is acting optimally in an agent-experiment loop (Fig. 1 B) according to the model of Section 2, and want to infer properties of the agent's perceptual and action processes, which are represented by parameters $\boldsymbol{\theta}$. In the examples in this paper, we have treated all matrices except the subjective control costs ($R$) and parameters of the task objective ($Q$) as given. This choice is motivated by the fact that the cost function is usually the least understood quantity in a behavioral experiment, while sensorimotor researchers often have quite accurate models for the dynamics in the tasks they are studying and for subjects' noise characteristics. In principle, however, our probabilistic formulation of the inverse optimal control problem allows inferring parameters $\boldsymbol{\theta}$ of any of the matrices of the system by evaluating the likelihood function w.r.t. those parameters. Given a set of $N$ independent trajectories $\{\boldsymbol{x}_{1:T}\}^{i=1:N}$, each of length $T$, we can infer $\boldsymbol{\theta}$ by maximizing the product of their likelihoods $p(\boldsymbol{x}_{1:T} \mid \boldsymbol{\theta})$, each decomposing as

$$p(\boldsymbol{x}_{1:T} \mid \boldsymbol{\theta}) = p_{\boldsymbol{\theta}}(\boldsymbol{x}_1) \prod_{t=1}^{T-1} p_{\boldsymbol{\theta}}(\boldsymbol{x}_{t+1} \mid \boldsymbol{x}_{1:t}). \tag{6}$$

In the following, we drop the explicit dependency of the parameters $\boldsymbol{\theta}$. The graphical model from the agent's point of view (Fig. 1 A) is structurally identical to that from the experimenter's perspective (Fig. 1 C). But, since we as experimenter observe the true states $\boldsymbol{x}_{1:T}$ instead of the agent's noisy observations $\boldsymbol{y}_{1:T}$, the usual Markov property does not hold and each $\boldsymbol{x}_t$ generally depends on all previous states $\boldsymbol{x}_{1:t-1}$ via the agent's estimates and actions. To efficiently compute the likelihood factors $p(\boldsymbol{x}_t \mid \boldsymbol{x}_{1:t-1})$, we track our belief about the agent's belief $p(\tilde{\boldsymbol{x}}_t \mid \boldsymbol{x}_{1:t})$, which gives a sufficient statistic for the history. This approach allows propagating our uncertainty about the agent's beliefs and actions over time and estimating the agent's belief.

To compute the likelihood function for some value of $\boldsymbol{\theta}$, we first determine the control and filter gains $L_t$ and $K_t$ using the iterative method introduced by Todorov [7]. We then compute an approximate likelihood factor $p(\boldsymbol{x}_t \mid \boldsymbol{x}_{1:t-1})$ for each time step in the following way (see Algorithm 1):

First, we determine the distribution $p(\boldsymbol{x}_{t+1}, \tilde{\boldsymbol{x}}_{t+1} \mid \boldsymbol{x}_t, \tilde{\boldsymbol{x}}_t)$, which describes the joint evolution of $\boldsymbol{x}_t$ and $\tilde{\boldsymbol{x}}_t$ (Section 3.1). Second, we combine it with the belief distribution $p(\tilde{\boldsymbol{x}}_t \mid \boldsymbol{x}_{1:t})$, yielding $p(\boldsymbol{x}_{t+1}, \tilde{\boldsymbol{x}}_{t+1} \mid \boldsymbol{x}_{1:t})$ (Section 3.2). As this step cannot be done in closed-form due to the signal-dependent noise, we introduce a Gaussian approximation of this quantity. Third, marginalizing over $\tilde{\boldsymbol{x}}_{t+1}$ gives the desired likelihood factor $p(\boldsymbol{x}_{t+1} \mid \boldsymbol{x}_{1:t})$, while conditioning on the observed true states $\boldsymbol{x}_{t+1}$ gives the statistic of the history $p(\tilde{\boldsymbol{x}}_{t+1} \mid \boldsymbol{x}_{1:t+1})$, which we use for computing the likelihood factor of the following time step.

In Section 3.3, we extend this procedure to the setting where the state $\boldsymbol{x}_t$ is only partially observed as a noisy linearly transformed version $\boldsymbol{o}_t$ (Fig. 1 D).

**Algorithm 1:** Approximate Likelihood Computation

---

**Result:** Approximate log likelihood of parameters $\boldsymbol{\theta}$
**Input:** Parameters $\boldsymbol{\theta}$, Data $\{\boldsymbol{x}_{1:T}^i\}_{i=1:N}$, Model
$L_t, K_t \leftarrow$ Approximate optimal controller and filter using the method of Todorov [7];
initialize $p(\tilde{\boldsymbol{x}}_0 \mid \boldsymbol{x}_0)$ as the experimenter's initial belief of the agent's belief;
**for each** trajectory $\boldsymbol{x}_{1:T}$ from $\{\boldsymbol{x}_{1:T}^i\}_{i=1:N}$ **do**
    **for** $t \leftarrow 0$ **to** $T-1$ **do**
        Compute $p(\boldsymbol{x}_{t+1}, \tilde{\boldsymbol{x}}_{t+1} \mid \boldsymbol{x}_{1:t})$ using Eq. (10);
        Marginalize over $\tilde{\boldsymbol{x}}_{t+1}$ to get $p(\boldsymbol{x}_{t+1} \mid \boldsymbol{x}_{1:t})$;
        Condition on $\boldsymbol{x}_{t+1}$ to get $p(\tilde{\boldsymbol{x}}_{t+1} \mid \boldsymbol{x}_{1:t+1})$ using Eq. (11);
    **end**
**end**
**return** $\sum_{i=1}^N \log p(\boldsymbol{x}_1^i) + \sum_{t=2}^T \log p(\boldsymbol{x}_t^i \mid \boldsymbol{x}_{1:t-1}^i)$

---

## 3.1 Joint dynamics of states and estimates

In this section, we derive the joint dynamics of states and estimates, specifying the distribution $p(\boldsymbol{x}_{t+1}, \tilde{\boldsymbol{x}}_{t+1} \mid \boldsymbol{x}_t, \tilde{\boldsymbol{x}}_t)$. To do so, we build on work by Van Den Berg et al. [52], who introduced this idea for the standard LQG case in the context of planning, and extend it to the model with state- and action-dependent noises as considered in Section 2. First, we substitute the control in the state update (1) with its law (5), giving

$$\boldsymbol{x}_{t+1} = A\boldsymbol{x}_t - BL_t\tilde{\boldsymbol{x}}_t + V\boldsymbol{\xi}_t - \sum_{i=1}^d \varepsilon_t^i C_i L_t \tilde{\boldsymbol{x}}_t, \tag{7}$$

and rewrite the filter update equation (4) as

$$\tilde{\boldsymbol{x}}_{t+1} = (A - BL_t)\tilde{\boldsymbol{x}}_t + K_t(\boldsymbol{y}_t - H\tilde{\boldsymbol{x}}_t) + E\boldsymbol{\eta}_t$$

$$= (A - BL_t - K_tH)\tilde{\boldsymbol{x}}_t + K_tH\boldsymbol{x}_t + K_tW\boldsymbol{\omega}_t + K_t\sum_{i=1}^c \epsilon_t^i D_i \boldsymbol{x}_t + E\boldsymbol{\eta}_t. \tag{8}$$

In the last equation, we have again inserted the control law (5), then the observation model (2), and rearranged terms. Equations (7) and (8) give us a representation of $\boldsymbol{x}_t$ and $\tilde{\boldsymbol{x}}_t$ which only depends on states or estimates from the previous time step. Stacking both equations together specifies the distribution $p(\boldsymbol{x}_{t+1}, \tilde{\boldsymbol{x}}_{t+1} \mid \boldsymbol{x}_t, \tilde{\boldsymbol{x}}_t)$, with

$$\begin{bmatrix} \boldsymbol{x}_{t+1} \\ \tilde{\boldsymbol{x}}_{t+1} \end{bmatrix} = \begin{bmatrix} A & -(B + \sum_{i=1}^d \varepsilon_t^i C_i)L_t \\ K_t(H + \sum_{i=1}^c \epsilon_t^i D_i) & A - BL_t - K_tH \end{bmatrix} \begin{bmatrix} \boldsymbol{x}_t \\ \tilde{\boldsymbol{x}}_t \end{bmatrix} + \begin{bmatrix} V & 0 & 0 \\ 0 & K_tW & E \end{bmatrix} \begin{bmatrix} \boldsymbol{\xi}_t \\ \boldsymbol{\omega}_t \\ \boldsymbol{\eta}_t \end{bmatrix}$$

$$=: (F_t + M_t \sum_{i=1}^c \epsilon_t^i D_i)\boldsymbol{x}_t + (\tilde{F}_t + \sum_{i=1}^d \varepsilon_t^i C_i \tilde{M}_t)\tilde{\boldsymbol{x}}_t + G_t\boldsymbol{\zeta}_t, \tag{9}$$

where $\boldsymbol{\zeta}_t \sim \mathcal{N}(0, I)$. For a detailed definition of the matrices $F_t, \tilde{F}_t, M_t, \tilde{M}_t$ see Appendix C.1.

## 3.2 Approximate propagation

We obtain the distribution $p(\boldsymbol{x}_{t+1}, \tilde{\boldsymbol{x}}_{t+1} \mid \boldsymbol{x}_{1:t})$ by propagating $p(\tilde{\boldsymbol{x}}_t \mid \boldsymbol{x}_{1:t})$ through the joint dynamics model (9). But, since the latter involves a product of Gaussian random variables $\varepsilon_t$ and $\tilde{\boldsymbol{x}}_t$, the resulting distribution is no longer Gaussian. To make likelihood computation tractable, we approximate it by a Gaussian using moment matching. This allows us to maintain an approximate Gaussian belief about the agent's belief and gives us an approximation of the likelihood function in Eq. (6).

First, we assume that our belief of the agent's belief at time step $t$ is given by a Gaussian distribution $p(\tilde{\boldsymbol{x}}_t \mid \boldsymbol{x}_{1:t}) = \mathcal{N}\left(\boldsymbol{\mu}_{\tilde{x}|x}, \Sigma_{\tilde{x}|x}\right)$. To approximately propagate $p(\tilde{\boldsymbol{x}}_t \mid \boldsymbol{x}_{1:t})$ and the observation $\boldsymbol{x}_t$ through Eq. (9), we compute the mean and variance of the resulting distribution via moment matching (see Appendix E) and obtain the approximation

$$p(\boldsymbol{x}_{t+1}, \tilde{\boldsymbol{x}}_{t+1} \mid \boldsymbol{x}_{1:t}) \approx \mathcal{N}\left( \begin{bmatrix} \boldsymbol{x}_{t+1} \\ \tilde{\boldsymbol{x}}_{t+1} \end{bmatrix} \middle| \hat{\boldsymbol{\mu}}_{t+1} = \begin{bmatrix} \hat{\boldsymbol{\mu}}_x \\ \hat{\boldsymbol{\mu}}_{\tilde{x}} \end{bmatrix}, \hat{\Sigma}_{t+1} = \begin{bmatrix} \hat{\Sigma}_{xx} & \hat{\Sigma}_{x\tilde{x}} \\ \hat{\Sigma}_{\tilde{x}x} & \hat{\Sigma}_{\tilde{x}\tilde{x}} \end{bmatrix} \right), \tag{10}$$

with

$$\hat{\boldsymbol{\mu}}_{t+1} = F_t \boldsymbol{x}_t + \tilde{F}_t \boldsymbol{\mu}_{\tilde{x}|x},$$

$$\hat{\Sigma}_{t+1} = M_t \Big( \sum_{i=1}^{c} D_i \boldsymbol{x}_t \boldsymbol{x}_t^T D_i^T \Big) M_t^T + \sum_{i=1}^{d} C_i \tilde{M}_t (\Sigma_{\tilde{x}|x} + \boldsymbol{\mu}_{\tilde{x}|x} \boldsymbol{\mu}_{\tilde{x}|x}^T) \tilde{M}_t^T C_i^T + \tilde{F}_t \Sigma_{\tilde{x}|x} \tilde{F}_t^T + G G^T.$$

Marginalizing over $\tilde{\boldsymbol{x}}_{t+1}$ gives an approximation of the likelihood factor of time step $t + 1$, $p(\boldsymbol{x}_{t+1} \mid \boldsymbol{x}_{1:t}) \approx \mathcal{N}(\hat{\boldsymbol{\mu}}_x, \hat{\Sigma}_{xx})$. On the other hand, conditioning on observation $\boldsymbol{x}_{t+1}$ gives the belief of the agent's belief for the following time step, $p(\tilde{\boldsymbol{x}}_{t+1} \mid \boldsymbol{x}_{1:t+1}) = \mathcal{N}\left( \hat{\boldsymbol{\mu}}_{\tilde{x}|x}, \hat{\Sigma}_{\tilde{x}|x} \right)$, with

$$\hat{\boldsymbol{\mu}}_{\tilde{x}|x} = \hat{\boldsymbol{\mu}}_{\tilde{x}} + \hat{\Sigma}_{\tilde{x}x} \hat{\Sigma}_{xx}^{-1} (\boldsymbol{x}_{t+1} - \hat{\boldsymbol{\mu}}_x), \qquad \hat{\Sigma}_{\tilde{x}|x} = \hat{\Sigma}_{\tilde{x}\tilde{x}} - \hat{\Sigma}_{\tilde{x}x} \hat{\Sigma}_{xx}^{-1} \hat{\Sigma}_{x\tilde{x}}. \tag{11}$$

We initialize $p(\tilde{\boldsymbol{x}}_0 \mid \boldsymbol{x}_0)$ with the initial belief of the agent.

### 3.3 Partial observability from the observer's point of view

In practice, we often do not have access to the full state $\boldsymbol{x}_t$ in the model, e.g., if there are unmeasured quantities of the physical world such as velocity and acceleration when using a tracking system which only provides measurements of position in time, or if we have latent variables in our model representing internal states of the observed agent. Furthermore, measurements might be noisy, e.g., due to the use of imprecise tracking hardware. We therefore consider the case where the state is partially observable for both the agent, i.e., the subject in the experiment, and the observer, i.e., the experimenter. In this case, we assume that the experimenter observes a linear transformation $\boldsymbol{o} \in \mathbb{R}^s$ of the state $\boldsymbol{x}_t$ with additive Gaussian noise, i.e.,

$$\boldsymbol{o}_t = S \boldsymbol{x}_t + U \boldsymbol{\vartheta}_t, \tag{12}$$

where $\boldsymbol{\vartheta}_t$ is a standard Gaussian random vector, resulting in the distribution $p(\boldsymbol{o}_t \mid \boldsymbol{x}_t) = \mathcal{N}\left( S \boldsymbol{x}_t, U U^T \right)$. The resulting Bayesian network is shown in Fig. 1 D. We can again formulate a joint dynamical system of $\boldsymbol{x}_t$ and $\tilde{\boldsymbol{x}}_t$ with additional observations $\boldsymbol{o}_t$, resulting in

$$\begin{bmatrix} \boldsymbol{x}_{t+1} \\ \tilde{\boldsymbol{x}}_{t+1} \\ \boldsymbol{o}_{t+1} \end{bmatrix} = \begin{bmatrix} A & -(B + \sum_{i=1}^{d} \varepsilon_t^i C_i) L_t \\ K_t(H + \sum_{i=1}^{c} \epsilon_t^i D_i) & A - B L_t - K_t H \\ SA & -S(B + \sum_{i=1}^{d} \varepsilon_t^i C_i) L_t \end{bmatrix} \begin{bmatrix} \boldsymbol{x}_t \\ \tilde{\boldsymbol{x}}_t \end{bmatrix} + \begin{bmatrix} V & 0 & 0 & 0 \\ 0 & K_t W & E & 0 \\ SV & 0 & 0 & U \end{bmatrix} \begin{bmatrix} \boldsymbol{\xi}_t \\ \boldsymbol{\omega}_t \\ \boldsymbol{\eta}_t \\ \boldsymbol{\vartheta} \end{bmatrix}$$

$$=: (F_t + M_t \sum_{i=1}^{c} \epsilon_t^i D_i) \boldsymbol{x}_t + (\tilde{F}_t + \sum_{i=1}^{d} \varepsilon_t^i C_i \tilde{M}_t) \tilde{\boldsymbol{x}}_t + G_t \boldsymbol{\zeta}_t, \tag{13}$$

with matrices $F_t, \tilde{F}_t, M_t, \tilde{M}_t$ defined accordingly (for definitions see Appendix C.2). Note that this equation is structurally the same as for the fully-observable case (Eq. (9)) and we have overloaded the matrix definitions to highlight that both can be treated similarly.

The likelihood of an observed trajectory decomposes as $p(\boldsymbol{o}_{1:T} \mid \boldsymbol{\theta}) = p_{\boldsymbol{\theta}}(\boldsymbol{o}_1) \prod_{t=2}^{T} p_{\boldsymbol{\theta}}(\boldsymbol{o}_t \mid \boldsymbol{o}_{1:t-1})$. For computing the factors $p(\boldsymbol{o}_t \mid \boldsymbol{o}_{1:t-1})$, we follow structurally the same steps as for the fully-observable case, but now $p(\boldsymbol{x}_t, \tilde{\boldsymbol{x}}_t \mid \boldsymbol{o}_{1:t})$ serves as sufficient statistic of the history: We first assume the distribution $p(\boldsymbol{x}_t, \tilde{\boldsymbol{x}}_t \mid \boldsymbol{o}_{1:t})$ to be Gaussian distributed and approximately propagate it through the joint dynamics model $p(\boldsymbol{x}_{t+1}, \tilde{\boldsymbol{x}}_{t+1}, \boldsymbol{o}_{t+1} \mid \boldsymbol{x}_t, \tilde{\boldsymbol{x}}_t)$ (Eq. (13)) by computing the mean and variance. We marginalize the resulting Gaussian approximation of $p(\boldsymbol{x}_{t+1}, \tilde{\boldsymbol{x}}_{t+1}, \boldsymbol{o}_{t+1} \mid \boldsymbol{o}_{1:t})$ over $\boldsymbol{x}_{t+1}, \tilde{\boldsymbol{x}}_{t+1}$, yielding the likelihood factors $p(\boldsymbol{o}_{t+1} \mid \boldsymbol{o}_{1:t})$. On the other hand, conditioning on the observation $\boldsymbol{o}_t$ gives the history statistic $p(\boldsymbol{x}_t, \tilde{\boldsymbol{x}}_t \mid \boldsymbol{o}_{1:t})$ for the following time step. All steps are very similar to the fully-observable case, but a more detailed description is given in Appendix D.

### 3.4 Parameter inference

In the previous sections, we have provided an algorithm for computing an approximate likelihood of the parameters $\boldsymbol{\theta}$ given a set of observed trajectories $\{\boldsymbol{x}_{1:T}^i\}_{i=1:N}$. To determine the optimal parameters, we maximize the likelihood, giving us a point estimate $\boldsymbol{\theta}_{\text{MLE}}$ of the true parameters. As one has to solve the control problem (determining $L_t$ and $K_t$) by an iterative procedure [7] for every

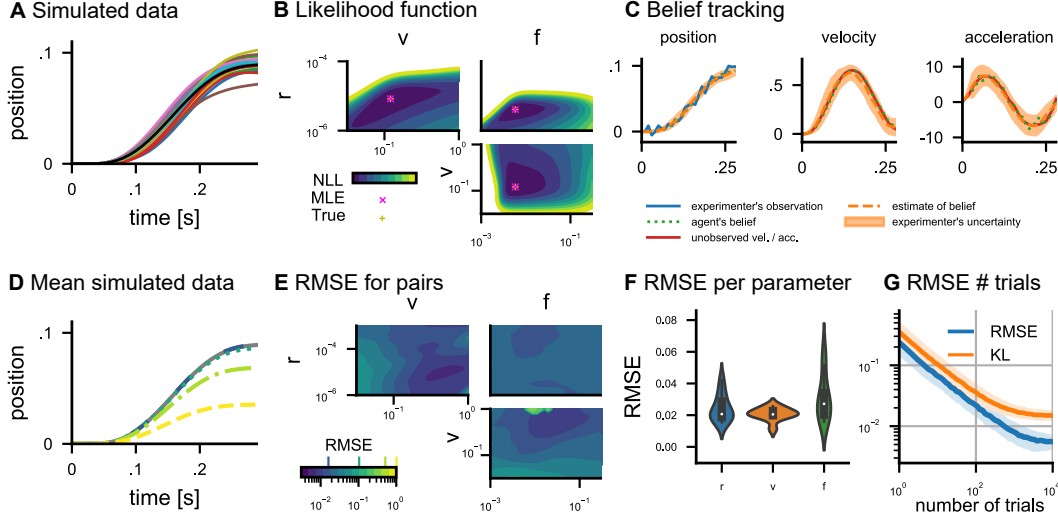

Figure 2: **Validation on synthetic reaching data. A** Simulated trajectories of the reaching task. **B** Negative log likelihood for different combinations of the parameters given data and the resp. third parameter. **C** Belief tracking: During a reaching movement, the agent has a belief about the position, velocity, and acceleration (green dotted lines). The experimenter observes a noisy version of the agent's actual movements (blue) and computes an estimate of the agent's belief (orange dashed lines with shaded region representing $2 \times$ SD). **D** Mean trajectories with RMSE 0.016 (MLE), 0.1, 0.5, and 1.0. **E** RMSE of the MLEs for all parameters on pairwise grids (for one value respective third parameter). The colors are the same as in the previous subplot. **F** Distribution over 10 repetitions of RMSE for each parameter across different values for the other parameters. **G** RMSE and KL divergences (between empirical distributions of true trajectories and simulated ones based on the MLE) with 0.2 and 0.8 quantiles for different numbers of trajectories with parameters as in (A).

likelihood evaluation, computing gradients of the likelihood (although possible) is not very efficient. We instead use the robust gradient-free optimizer BOBYQA[2] [53], which minimizes the negative log likelihood based on a quadratic approximation. Our implementation in `jax` [54] is available on github.[3]

## 4 Evaluation and applications

### 4.1 Validation on synthetic reaching data

We apply the introduced method to recover parameters in a single-joint reaching task with control-dependent noise and 5-dimensional state space (details in Appendix F.1). The goal is to bring the hand to a target while minimizing control effort. The cost function has three parameters: (i) $v$, the cost of the velocity at the final time step, (ii) $f$, the cost of the acceleration at the final timestep, (iii) $r$, the cost of actions at each timestep. Simulated data for the parameters $r = 10^{-5}, v = 0.2, f = 0.02$ are shown in Fig. 2 A. Visual inspection of the likelihood function (Fig. 2 B) shows that the maximum likelihood estimate (MLE) is very close to the true parameter values.

Once we have obtained the MLE, we can perform belief tracking, i.e., computing our approximate belief of the agent's belief $p(\tilde{\boldsymbol{x}}_t | \boldsymbol{o}_{1:t})$. As an example, we simulated $N = 20$ trajectories $\{\boldsymbol{o}_{1:T}\}^{i=1:N}$ with $T = 30$ from a partially observed version of the reaching task used above in which we only observe the position and treat velocity and acceleration as latent variables. Fig. 2 C shows our approximate belief about the agent's belief for the MLE parameters, together with the true agent's belief. Note that we can recover the agent's belief of state, velocity, and acceleration quite accurately from noisy observations of the position only.

---

[2]Python implementation under GNU GPL available at PyPI
[3]

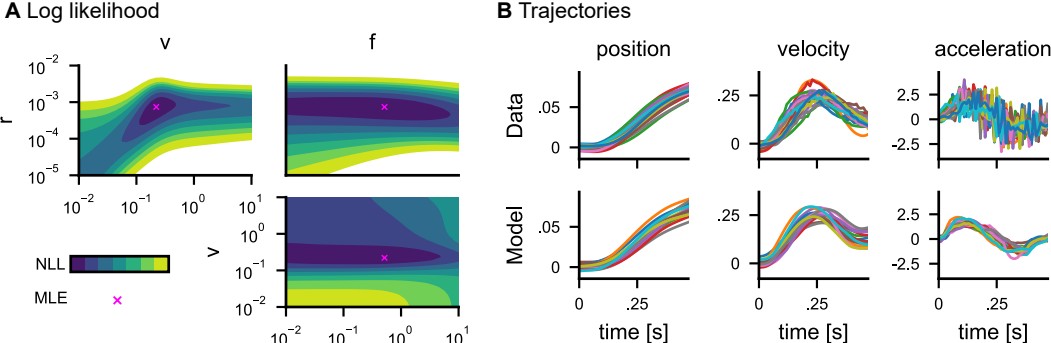

Figure 3: **Application to real reaching data. A** Negative log likelihoods for the three model parameters. **B** Reaching trajectories (velocities and accelerations computed with finite differences) and simulated data from the model with MLE parameters.

For the evaluation of parameter estimates, we compute their root mean squared errors (RMSEs) in logarithmic space. The effect of estimation errors on the resulting trajectories is illustrated in Fig. 2 D, where we simulated trajectories as in Fig. 2 A with different mean parameter errors. To show that our method yields good parameter estimates over a range of different parameter settings, we perform maximum likelihood estimations (MLEs) of all three parameters for different true parameter values of which two were chosen from a pairwise grid while the third one was left as in Fig. 2 A. In this analysis, we used 100 simulated trajectories and 10 repetitions each. In Fig. 2 E, which shows the resulting RMSEs for different combinations of true parameters on pairwise grids, we demonstrate that the RMSEs are small over a wide range of parameter values. The RMSEs (Fig. 2 F) across different values for the respective other two parameters and 10 repetitions were $2.4 \times 10^{-2}$ ($r$), $2.1 \times 10^{-2}$ ($v$) and $3.1 \times 10^{-2}$ ($f$). We compared the RMSEs obtained by our estimation method to the ones of two baseline approaches. A first baseline is obtained by running a version of the algorithm without signal-dependent noise (i.e., using the basic LQG during inference), for which the likelihood can be evaluated exactly in closed-form. The results on the reaching problem (Section 4.1) in terms of RMSE of the parameter estimates are worse by roughly two orders of magnitude (RMSE of 1.766 vs 0.027). A second, stronger baseline is obtained by setting the additive noise in the standard LQG to the average noise magnitude of simulated trajectories. In this case, the RMSEs are still worse by roughly an order of magnitude (RMSE of 0.702 vs 0.027). Note that the information on the average noise level would not be readily available for real data without knowing the true parameters and therefore constitutes a strong baseline. A plot visualizing the results of the baselines for each parameter is provided in Appendix G.1.1.

Finally, we investigated the influence of the number of samples by evaluating estimates of trajectories as in Fig. 2 A for different numbers of trials (Fig. 2 G). As expected, more trials increase the accuracy of the estimates and lead to lower error. We estimated the convergence rate by fitting a line to the log-log plot and obtained 0.78 for the RMSE. An analysis using the Kullback–Leibler divergence between the empirical trajectory distributions of the true and maximum likelihood parameters as an additional evaluation metric can be found in Appendix G.1.2.

We perform a similar analysis for generated partially observed trajectories in which we only observe the position and treat velocities, accelerations, and forces as latent variables. The results are qualitatively very similar and are therefore presented in Appendix G.1.3. An additional empirical evaluation of the impact of the moment matching approximation is given in Appendix G.1.4.

## 4.2 Application to real reaching movements

To show the applicability to real data, we apply our method to reaching trajectories from a previously published experiment [55], in which a rhesus monkey had to perform center-out reaching movements and hold its hand at the target to receive a reward. Since the data contains only measurements of position (velocity and acceleration are computed using finite differences), we use the partially observable version of the reaching model described in Section 4.1, treating velocity and acceleration as latent variables. The approximate likelihood functions with respective MLEs of the three parameters are shown in Fig. 3 A, indicating that we can determine the parameter set for which the trajectories are

most likely. Fig. 3 B shows the given trajectories together with simulations using our model with the MLE parameters. We observe that the inferred parameters produce simulated data that convincingly look like the real data and provides smooth estimates of the latent velocity and acceleration profiles.

## 4.3 Application to eye movements

We also apply our method to a model of saccadic eye movements which was presented by Crevecoeur and Kording [56]. This model captures fixating one's eyes to an initial point and then performing a saccade to fixate another point. A cost parameter $(r)$ is used to trade off the cost of the movement and the deviation from the target. As this model is an LQG model with control-dependent noises, it directly allows the application of our method for recovering the parameter. Fig. 4 A shows simulated trajectories representing typical eye movements encountered in the experiments. The MLEs based on simulated data for a range of 20 different parameter values (100 repetitions each) are shown in Fig. 4 B. Except for very few outliers, the estimated parameters are very close to the true parameters.

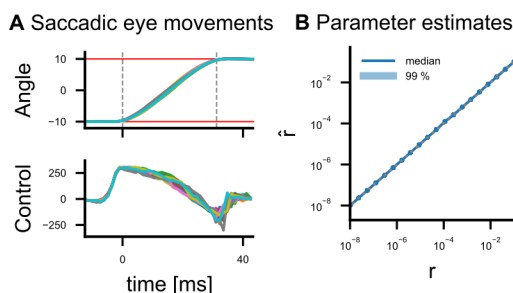

Figure 4: **Saccadic eye movements. A** Generated trajectories. **B** Median and quantiles of MLEs for a range of true parameter values of $r$.

## 4.4 Application to random problems

To demonstrate that the inference method works on a wide range of problems defined according to the model definition (Section 2), we evaluate it on randomly generated problems with 5-dimensional state-space and two-dimensional action space. Detailed information on the generation procedure is given in Appendix F.3. Each model has two parameters ($r_1$ and $r_2$) which again represent the cost of control effort in each of the two movement directions. For each model, we sampled a true value of parameter $r_2$ from a uniform random distribution and estimated both parameters jointly by maximizing the approximate likelihood. In Fig. 5 A we show the errors for a range of parameter values $r_1$ for different random models. The median and quantiles for the results of 2000 random problems are shown in Fig. 5 B. One can observe that the estimates are generally very close to the true parameters. The results for the other parameter $r_2$ are basically identical since the problem is symmetric w.r.t. the parameters, but we include the results in Appendix G.2.

## 5 Conclusion

In this paper, we investigated the inverse optimal control problem under signal-dependent noise. We formalized the problem as a POMDP and introduced a first method for inferring cost parameters of an agent in a linear-quadratic control problem with signal-dependent noise. Numerical simulations show

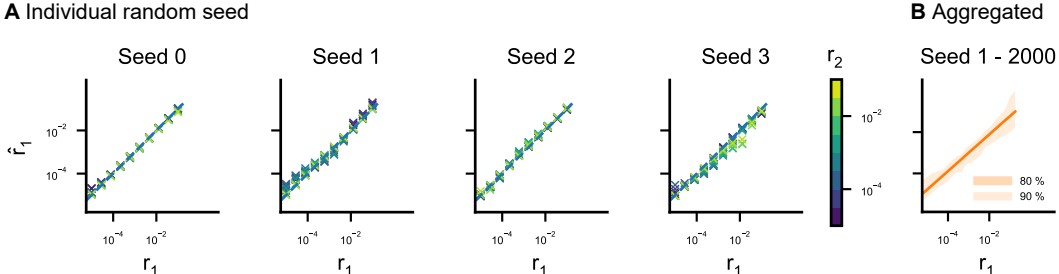

Figure 5: **Random problems. A** MLEs for a range of parameter values of $r_1$ for different random problems and different values of $r_2$ (color). **B** Aggregated results (median, percentiles) for 2000 random problems.

that accurate inference of cost parameters given synthetic data is feasible in random control problems, simulated arm movements, and simulated eye movements. Additionally, the method can be used to probabilistically infer the belief of the considered agent. Furthermore, the method was applied to real data from a macaque monkey performing reaching movements. The inferred parameters reproduce reaching data in simulation that convincingly agrees with the original data and provides smooth estimates of the latent velocity and acceleration profiles. More recent and more general methods for optimal control in high-dimensional continuous domains exist, but while some of these may provide interpretable non-linear features, they consider deterministic MDPs without noise and assume full observability [48], others relying on function approximation through neural networks including GANs, are useful in engineering applications but may not provide a computational level explanation of behavior [31, 32]. Taken together, our method does not require designing a cost function and testing for similarity of simulated trajectories with experimental data, but allows inferring the cost functions directly from behavior, thereby reconciling normative and descriptive approaches to human sensorimotor behavior.

**Limitations and Future work** The proposed algorithms are based on stochastic optimal feedback control with linear dynamics, quadratic cost functions, and signal-dependent noise. As such, the first limitation lies in the restriction to problems with linear dynamics. An extension to non-linear dynamics could be achieved by linearizing the dynamics locally or more generally by using a framework that iteratively linearizes the dynamics at each time step, e.g., iLQG [57]. Similarly, while quadratic cost functions allow modeling a wide range of costs and benefits within sensorimotor control, certain cost functions such as exponential discounting may be more cumbersome to accommodate.

While the presented method was able to recover cost functions in the considered problems, higher-dimensional parameter spaces will likely pose difficulties in finding unique point estimates of parameters. This problem could be addressed by using appropriate structured prior distributions over parameters. A fully Bayesian treatment could be realized by using Markov chain Monte Carlo involving our likelihood model. Further research should similarly investigate the limits of the Gaussian approximation of the likelihood. In the cases considered here, the approximation of the likelihood function appeared to be unbiased up to an RMSE of approximately $3 \times 10^{-2}$, however, approximating distributions by simpler ones may introduce systematic biases and noise. Possible extensions could resort to using particle filters albeit at a higher computational cost.

Another issue regarding higher-dimensional parameter spaces is that evaluation of the likelihood function becomes quite expensive by determining the optimal controller and optimal filter iteratively. Even if this procedure takes only one second on a common PC for the reaching task, optimization in high-dimensional spaces requires a larger number of function evaluations, which renders inference costly. By relying on the iterative procedure, it also becomes difficult to compute gradients, which may prevent the use of efficient gradient-based solvers. While for our applications a solver based on quadratic approximations was efficient, one could resort to Bayesian optimization.

Future work in the area of robotics may explore applying the inferred cost functions in the training of visuomotor policies [58, 59] in humanoid robots with reinforcement learning [60]. Possible applications include utilizing the inferred cost functions in apprenticeship learning [61, 62, 63], in which a policy is learned from demonstrations of a potentially suboptimal demonstrator or teacher. Similarly, applications may also include transfer learning [64, 65, 66], in which learned policies or cost functions are transferred to related tasks.

Finally, characterizing individual human subjects by analyzing their behavior may in principle be used with negative societal impact. In the context of scientific investigations of human sensorimotor control within cognitive science and neuroscience, only anonymized behavioral data for the understanding of the human mind and brain are employed.

# Acknowledgments and Disclosure of Funding

We acknowledge support from the Lichtenberg high performance computing facility of the TU Darmstadt, the projects 'The Adaptive Mind' and 'The Third Wave of AI' funded by the Excellence Program and the project 'WhiteBox' funded by the Priority Program LOEWE of the Hessian Ministry of Higher Education, Science, Research and Art.

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
