## A  Notation

Table 1 provides an overview of the notation used in this paper. We distinguish between variables used in the original model described in Section 2 [7] and quantities necessary for our inference method described in Section 3.

Table 1: Notation

| | |
|---|---|
| $\boldsymbol{x}_t \in \mathbb{R}^m$ | state at time $t$ |
| $\boldsymbol{u}_t \in \mathbb{R}^p$ | action |
| $\boldsymbol{y}_t \in \mathbb{R}^k$ | agent's observation of the state |
| $T$ | number of time steps |
| $A, B, H$ | system dynamics and agent's observation matrices |
| $\boldsymbol{\xi}_t, \boldsymbol{\omega}_t, \boldsymbol{\eta}_t, \epsilon_t, \varepsilon_t$ | standard Gaussian noise terms |
| $V, W, E$ | scaling matrices for system, observation and estimation noise |
| $C_1, \ldots C_d$ | scaling matrices for control-dependent system noise |
| $D_1, \ldots D_c$ | scaling matrices for state-dependent observation noise |
| $Q_t$ | state-dependent costs |
| $R_t$ | control-dependent costs |
| $\tilde{\boldsymbol{x}}_t$ | agent's state estimate |
| $K_t$ | filter gain matrices |
| $L_t$ | control gain matrices |
| $S$ | researcher's observation matrix |
| $U$ | scaling matrix for researcher's observation noise |
| $\boldsymbol{\theta}$ | model parameters |
| $N$ | number of trials |
| $F, \tilde{F}$ | dynamics of the joint dynamical system |
| $\boldsymbol{\zeta}_t, \boldsymbol{\vartheta}_t$ | standard Gaussian noise terms |
| $M, \tilde{M}$ | scaling of the signal-dependent noises in the joint dynamical system |
| $G$ | scaling of the signal-independent noise in the joint dynamical system |
| $\boldsymbol{o}_t \in \mathbb{R}^s$ | experimenter's observation of the state |
| $\hat{\boldsymbol{\mu}}_t, \hat{\Sigma}_t$ | mean and covariance of the Gaussian approximation of state and agent's estimate |
| $\hat{\boldsymbol{\mu}}_{\tilde{x}|x}, \hat{\Sigma}_{\tilde{x}|x}$ | mean and covariance of the experimenter's belief about the agent's estimate |

## B  Approximate optimal control of LQG systems with sensorimotor noise characteristics

For approximately solving a system as described in Section 2, the optimal filters $K_t$ and controllers $L_t$ can be iteratively determined in an alternating fashion, leaving the respective other one constant Todorov [7].

Given filter matrices $K_t$, the optimal control matrices $L_t$ are computed in form of a backward pass as

$$L_t = \left( R_t + B^T P_{t+1}^x B + \sum_i C_i^T \left( P_{t+1}^x + P_{t+1}^e \right) C_i \right)^{-1} B^T P_{t+1}^x A$$

$$P_t^x = Q_t + A^T P_{t+1}^x \left( A - BL_t \right) + \sum_i D_i^T K_t^T P_{t+1}^e K_t D_i$$

$$P_t^e = A^T P_{t+1}^x BL_t + \left( A - K_t H \right)^T P_{t+1}^e \left( A - K_t H \right)$$

$$s_t = \operatorname{tr} \left( P_{t+1}^x VV^T + P_{t+1}^e \left( VV^T + EE^T + K_t WW^T K_t^T \right) \right) + s_{t+1},$$

where we initialize $P_T^x = Q_T, P_T^e = 0, s_T = 0$.

Given optimal control matrices $L_t$, the optimal filter matrices $K_t$ are computed in form of a forward pass as

$$K_t = A\Sigma_t^e H^T \left( H\Sigma_t^e H^T + WW^T + \sum_i D_i \left( \Sigma_t^e + \Sigma_t^{\tilde{x}} + \Sigma_t^{\widetilde{ex}} + \Sigma_t^{\widetilde{xe}} \right) D_i^T \right)^{-1}$$

$$\Sigma_{t+1}^e = VV^T + EE^T + (A - K_t H)\Sigma_t^e A^T + \sum_i C_i L_t \Sigma_t^{\tilde{x}} L_t^T C_i^T$$

$$\Sigma_{t+1}^{\tilde{x}} = EE^T + K_t H \Sigma_t^e A^T + (A - BL_t)\Sigma_t^{\tilde{x}} (A - BL_t)^T$$
$$+ (A - BL_t)\Sigma_t^{\widetilde{xe}} H^T K_t^T + K_t H \Sigma_t^{\widetilde{ex}} (A - BL_t)^T$$

$$\Sigma_{t+1}^{\widetilde{xe}} = (A - BL_t)\Sigma_t^{\widetilde{xe}} (A - K_t H)^T - EE^T,$$

with $\Sigma_t^{\widetilde{ex}} = \left(\Sigma_t^{\widetilde{xe}}\right)^T$ and we initialize

$$\Sigma_1^e = \Sigma_1$$
$$\Sigma_1^{\tilde{x}} = \tilde{x}_1 \tilde{x}_1^T$$
$$\Sigma_1^{\widetilde{xe}} = 0.$$

## C Joint update equation derivation

### C.1 Fully-observable state

Stacking Eq. (7) and Eq. (8) into a vector, gives

$$\begin{bmatrix} x_{t+1} \\ \tilde{x}_{t+1} \end{bmatrix} = \begin{bmatrix} Ax_t - BL\tilde{x}_t + V\xi_t - \sum_{i=1}^d \varepsilon_t^i C_i L_t \tilde{x}_t, \\ (A - BL_t - K_t H)\tilde{x}_t + K_t H x_t + K_t W\omega_t + K_t \sum_{i=1}^c \epsilon_t^i D_i x_t + E\eta_t \end{bmatrix}$$

$$= \begin{bmatrix} A \\ K_t H \end{bmatrix} x_t + \begin{bmatrix} -BL_t \\ A - BL_t - K_t H \end{bmatrix} \tilde{x} + \begin{bmatrix} 0 \\ K_t \end{bmatrix} \sum_{i=1}^c \epsilon_t^i D_i x_t + \sum_{i=1}^d \varepsilon_t^i C_i \begin{bmatrix} -L_t \\ 0 \end{bmatrix} \tilde{x}$$

$$+ \begin{bmatrix} V & 0 & 0 \\ 0 & K_t W & E \end{bmatrix} \begin{bmatrix} \xi_t \\ \omega_t \\ \eta_t \end{bmatrix}$$

$$= \left( \begin{bmatrix} A \\ K_t H \end{bmatrix} + \begin{bmatrix} 0 \\ K_t \end{bmatrix} \sum_{i=1}^c \epsilon_t^i D_i \right) x_t + \left( \begin{bmatrix} -BL_t \\ A - BL_t - K_t H \end{bmatrix} + \sum_{i=1}^d \varepsilon_t^i C_i \begin{bmatrix} -L_t \\ 0 \end{bmatrix} \right) \tilde{x}$$

$$+ \begin{bmatrix} V & 0 & 0 \\ 0 & K_t W & E \end{bmatrix} \begin{bmatrix} \xi_t \\ \omega_t \\ \eta_t \end{bmatrix}$$

$$=: (F_t + M_t \sum_{i=1}^c \epsilon_t^i D_i)x_t + (\tilde{F}_t + \sum_{i=1}^d \varepsilon_t^i C_i \tilde{M}_t)\tilde{x} + G_t \zeta_t.$$

### C.2 Partially-observable state

Stacking Eq. (7), Eq. (8), and Eq. (12) into a vector, gives

$$\begin{bmatrix} x_{t+1} \\ \tilde{x}_{t+1} \end{bmatrix} = \begin{bmatrix} Ax_t - BL\tilde{x}_t + V\xi_t - \sum_{i=1}^d \varepsilon_t^i C_i L_t \tilde{x}_t \\ (A - BL_t - K_t H)\tilde{x}_t + K_t H x_t + K_t W\omega_t + K_t \sum_{i=1}^c \epsilon_t^i D_i x_t + E\eta_t \\ S\left( Ax_t - BL\tilde{x}_t + V\xi_t - \sum_{i=1}^d \varepsilon_t^i C_i L_t \tilde{x}_t \right) + U\vartheta_t \end{bmatrix}$$

$$= \begin{bmatrix} A \\ K_t H \\ SA \end{bmatrix} \boldsymbol{x}_t + \begin{bmatrix} -BL_t \\ A - BL_t - K_t H \\ -SBL_t \end{bmatrix} \tilde{\boldsymbol{x}} + \begin{bmatrix} 0 \\ K_t \\ 0 \end{bmatrix} \sum_{i=1}^{c} \epsilon_t^i D_i \boldsymbol{x}_t + \sum_{i=1}^{d} \varepsilon_t^i C_i \begin{bmatrix} -L_t \\ 0 \\ -SL_t \end{bmatrix} \tilde{\boldsymbol{x}}$$

$$+ \begin{bmatrix} V & 0 & 0 & 0 \\ 0 & K_t W & E & 0 \\ SV & 0 & 0 & U \end{bmatrix} \begin{bmatrix} \boldsymbol{\xi}_t \\ \boldsymbol{\omega}_t \\ \boldsymbol{\eta}_t \\ \boldsymbol{\vartheta} \end{bmatrix}$$

$$= \left( \begin{bmatrix} A \\ K_t H \\ SA \end{bmatrix} + \begin{bmatrix} 0 \\ K_t \\ 0 \end{bmatrix} \sum_{i=1}^{c} \epsilon_t^i D_i \right) \boldsymbol{x}_t + \left( \begin{bmatrix} -BL_t \\ A - BL_t - K_t H \\ -SBL_t \end{bmatrix} + \sum_{i=1}^{d} \varepsilon_t^i C_i \begin{bmatrix} -L_t \\ 0 \\ -SL_t \end{bmatrix} \right) \tilde{\boldsymbol{x}}$$

$$+ \begin{bmatrix} V & 0 & 0 & 0 \\ 0 & K_t W & E & 0 \\ SV & 0 & 0 & U \end{bmatrix} \begin{bmatrix} \boldsymbol{\xi}_t \\ \boldsymbol{\omega}_t \\ \boldsymbol{\eta}_t \\ \boldsymbol{\vartheta} \end{bmatrix}$$

$$=: (F_t + M_t \sum_{i=1}^{c} \epsilon_t^i D_i) \boldsymbol{x}_t + (\tilde{F}_t + \sum_{i=1}^{d} \varepsilon_t^i C_i \tilde{M}_t) \tilde{\boldsymbol{x}} + G_t \boldsymbol{\zeta}_t.$$

## D   Approximate likelihood in case of partially-observable state

In the following, we define $\breve{\boldsymbol{x}}_t$ as the true state $\boldsymbol{x}_t$ and the agent's belief $\tilde{\boldsymbol{x}}_t$ stacked together, i.e., $\breve{\boldsymbol{x}}_t = [\boldsymbol{x}_t, \tilde{\boldsymbol{x}}_t]$. First, we assume that the belief of the agent's belief at time step $t$ is given by a Gaussian distribution

$$p(\boldsymbol{x}_t, \tilde{\boldsymbol{x}}_t \mid \boldsymbol{o}_{1:t}) = \mathcal{N} \left( \begin{bmatrix} \boldsymbol{x}_t \\ \tilde{\boldsymbol{x}}_t \end{bmatrix} \middle| \boldsymbol{\mu}_{\breve{x}|o} = \begin{bmatrix} \boldsymbol{\mu}_{x|o} \\ \boldsymbol{\mu}_{\tilde{x}|o} \end{bmatrix}, \Sigma_{\breve{x}|o} = \begin{bmatrix} \Sigma_{x|o} & \Sigma_{x\tilde{x}|o} \\ \Sigma_{\tilde{x}x|o} & \Sigma_{\tilde{x}|o} \end{bmatrix} \right).$$

To approximately propagate $p(\boldsymbol{x}_t, \tilde{\boldsymbol{x}}_t \mid \boldsymbol{o}_{1:t})$ through Eq. (9), we compute the mean and variance of the resulting distribution via moment matching (see Appendix E) and obtain the approximation

$$p(\breve{\boldsymbol{x}}_{t+1}, \boldsymbol{o}_{t+1} \mid \boldsymbol{o}_{1:t}) \approx \mathcal{N} \left( \begin{bmatrix} \breve{\boldsymbol{x}}_{t+1} \\ \boldsymbol{o}_{t+1} \end{bmatrix} \middle| \hat{\boldsymbol{\mu}}_{t+1} = \begin{bmatrix} \boldsymbol{\mu}_{\breve{x}} \\ \hat{\boldsymbol{\mu}}_o \end{bmatrix}, \hat{\Sigma}_{t+1} = \begin{bmatrix} \hat{\Sigma}_{\breve{x}} & \hat{\Sigma}_{\breve{x}o} \\ \hat{\Sigma}_{o\breve{x}} & \hat{\Sigma}_o \end{bmatrix} \right), \tag{14}$$

with

$$\hat{\boldsymbol{\mu}}_{t+1} = F_t \boldsymbol{\mu}_{x|o} + \tilde{F}_t \boldsymbol{\mu}_{\tilde{x}|o} = \breve{F} \boldsymbol{\mu}_{\breve{x}|o},$$

$$\hat{\Sigma}_{t+1} = M_t \left( \sum_{i=1}^{c} D_i (\Sigma_{x|o} + \boldsymbol{\mu}_{x|o} \boldsymbol{\mu}_{x|o}^T) D_i^T \right) M^T + \sum_{i=1}^{d} C_i \tilde{M} (\Sigma_{\tilde{x}|o} + \boldsymbol{\mu}_{\tilde{x}|o} \boldsymbol{\mu}_{\tilde{x}|o}^T) \tilde{M}^T C_i^T$$
$$+ \breve{F} \Sigma_{\breve{x}|o} \breve{F}^T + GG^T,$$

where $\breve{F}_t$ consists of $F_t$ and $\tilde{F}_t$ vertically stacked, i.e., $\breve{F}_t := \begin{bmatrix} F_t^T & \tilde{F}_t^T \end{bmatrix}^T$.

Marginalizing over $\boldsymbol{x}_{t+1}$ and $\tilde{\boldsymbol{x}}_{t+1}$ gives an approximation of the likelihood factor of time step $t + 1$, $p(\boldsymbol{o}_{t+1} \mid \boldsymbol{o}_{1:t}) \approx \mathcal{N}(\hat{\boldsymbol{\mu}}_o, \hat{\Sigma}_{oo})$. On the other hand, conditioning on observation $\boldsymbol{o}_{t+1}$ gives the belief of the state and the agent's estimate for the following time step, $p(\boldsymbol{x}_{t+1}, \tilde{\boldsymbol{x}}_{t+1} \mid \boldsymbol{o}_{1:t+1}) = \mathcal{N} \left( \hat{\boldsymbol{\mu}}_{\breve{x}|o}, \hat{\Sigma}_{\breve{x}|o} \right)$, with

$$\hat{\boldsymbol{\mu}}_{\breve{x}|o} = \hat{\boldsymbol{\mu}}_{\breve{x}} + \hat{\Sigma}_{\breve{x}o} \hat{\Sigma}_{oo}^{-1} (\boldsymbol{o}_{t+1} - \hat{\boldsymbol{\mu}}_o), \qquad \hat{\Sigma}_{\breve{x}|o} = \hat{\Sigma}_{\breve{x}\breve{x}} - \hat{\Sigma}_{\breve{x}o} \hat{\Sigma}_{oo}^{-1} \hat{\Sigma}_{o\breve{x}}. \tag{15}$$

We initialize $p(\boldsymbol{x}_0, \tilde{\boldsymbol{x}}_0 \mid \boldsymbol{o}_0)$ with our initial belief of the state and of the agent's belief.

## E   Derivation of approximate propagation

For a fully-observable state, the goal is to derive a closed-form approximation for $p(\boldsymbol{x}_{t+1}, \tilde{\boldsymbol{x}}_{t+1} \mid \boldsymbol{x}_{1:t})$ when propagating the (approximate) belief $p(\tilde{\boldsymbol{x}}_t \mid \boldsymbol{x}_{1:t})$ and the state $\boldsymbol{x}_t$ through the extended dynamics

model $p(\boldsymbol{x}_{t+1}, \tilde{\boldsymbol{x}}_{t+1} \mid \boldsymbol{x}_t, \tilde{\boldsymbol{x}}_t)$ by matching the result with a Gaussian distribution. In this section we will consider first the more general case where the state is partially observable, and derive the equations for a fully-observable state as a special case afterwards. As the fully- and partially-observable cases differ in the number of random variables (for partial-observability we have random variables $\boldsymbol{o}_t$ in addition to the true state $\boldsymbol{x}_t$), we will consider a general joint dynamics model $p(\boldsymbol{w}_{t+1} \mid \boldsymbol{x}_t, \tilde{\boldsymbol{x}}_t)$ and general observations $\boldsymbol{o}_t$. Then, the goal becomes to derive the approximation for $p(\boldsymbol{w}_{t+1} \mid \boldsymbol{o}_{1:t})$ when propagating the belief $p(\boldsymbol{x}_t, \tilde{\boldsymbol{x}}_t \mid \boldsymbol{o}_{1:t})$ through the model $p(\boldsymbol{w}_{t+1} \mid \boldsymbol{x}_t, \tilde{\boldsymbol{x}}_t)$.

We restate the update equation for $\boldsymbol{w}_{t+1}$, coinciding with both Eq. (9) and Eq. (13),

$$\boldsymbol{w}_{t+1} = (F_t + M_t \sum_{i=1}^{c} \epsilon_t^i D_i)\boldsymbol{x}_t + (\tilde{F}_t + \sum_{i=1}^{d} \varepsilon_t^i C_i \tilde{M}_t)\tilde{\boldsymbol{x}} + G_t \boldsymbol{\zeta}_t, \tag{16}$$

where the matrices $F_t, M_t, \tilde{F}_t, C_t,$ and $G_t$ are partitioned as described in Appendix C.1.

We assume that $p(\boldsymbol{x}_t, \tilde{\boldsymbol{x}}_t \mid \boldsymbol{o}_{1:t})$ follows a Gaussian distribution with

$$p(\boldsymbol{x}_t, \tilde{\boldsymbol{x}}_t \mid \boldsymbol{o}_{1:t}) \sim \mathcal{N}\left( \begin{bmatrix} \boldsymbol{x}_t \\ \tilde{\boldsymbol{x}}_t \end{bmatrix} \middle| \; \boldsymbol{\mu}_t = \begin{bmatrix} \boldsymbol{\mu}_x \\ \boldsymbol{\mu}_{\tilde{x}} \end{bmatrix}, \Sigma_t = \begin{bmatrix} \Sigma_{xx} & \Sigma_{x\tilde{x}} \\ \Sigma_{\tilde{x}x} & \Sigma_{\tilde{x}\tilde{x}} \end{bmatrix} \right). \tag{17}$$

To match $p(\boldsymbol{w}_{t+1} \mid \boldsymbol{o}_{1:t})$ with a Gaussian distribution, we compute the mean $\hat{\boldsymbol{\mu}}$ and variance $\hat{\Sigma}$ of Eq. (16) where $\boldsymbol{x}$ and $\tilde{\boldsymbol{x}}$ are distributed according to Eq. (17). In the following, we will drop time indices to enhance readability. For the mean, we obtain

$$
\begin{aligned}
\hat{\boldsymbol{\mu}} &= \mathbb{E}_{\boldsymbol{x},\boldsymbol{\epsilon},\tilde{\boldsymbol{x}},\boldsymbol{\varepsilon},\boldsymbol{\zeta}} \left[ (F + M \sum_{i=1}^{c} D_i \epsilon^i)\boldsymbol{x} + (\tilde{F} + \sum_{i=1}^{d} C_i \varepsilon^i \tilde{M})\tilde{\boldsymbol{x}} + G\boldsymbol{\zeta} \right] \\
&= \mathbb{E}_{\boldsymbol{x},\boldsymbol{\epsilon}} \left[ (F + M \sum_{i=1}^{c} D_i \epsilon^i)\boldsymbol{x} \right] + \mathbb{E}_{\tilde{\boldsymbol{x}},\boldsymbol{\varepsilon}} \left[ (\tilde{F} + \sum_{i=1}^{d} C_i \varepsilon^i \tilde{M})\tilde{\boldsymbol{x}} \right] + \mathbb{E}_{\boldsymbol{\zeta}} \left[ G\boldsymbol{\zeta} \right] \\
&= \mathbb{E}_{\boldsymbol{x}} \left[ (F + M \sum_{i=1}^{c} D_i \, \mathbb{E}_{\epsilon^i}\left[\epsilon^i\right])\boldsymbol{x} \right] + \mathbb{E}_{\tilde{\boldsymbol{x}}} \left[ (\tilde{F} + \sum_{i=1}^{d} C_i \, \mathbb{E}_{\varepsilon^i}\left[\varepsilon^i\right] \tilde{M})\tilde{\boldsymbol{x}} \right] + G \, \mathbb{E}_{\boldsymbol{\zeta}} \left[ \boldsymbol{\zeta} \right] \\
&= \mathbb{E}_{\boldsymbol{x}} \left[ F\boldsymbol{x} \right] + \mathbb{E}_{\tilde{\boldsymbol{x}}} \left[ \tilde{F}\tilde{\boldsymbol{x}} \right] + G \, \mathbb{E}_{\boldsymbol{\zeta}} \left[ \boldsymbol{\zeta} \right] \\
&= F\boldsymbol{\mu}_x + \tilde{F}\boldsymbol{\mu}_x \\
&=: \hat{\boldsymbol{\mu}}_x + \hat{\boldsymbol{\mu}}_x.
\end{aligned}
$$

For the variance, we use that $\boldsymbol{\zeta}$ is independent of $\breve{\boldsymbol{x}} := \left[ \boldsymbol{x}^T, \, \tilde{\boldsymbol{x}}^T \right]^T$, therefore

$$\hat{\Sigma} = \hat{\Sigma}_{\breve{x}} + \hat{\Sigma}_{\boldsymbol{\zeta}},$$

with $\hat{\Sigma}_{\boldsymbol{\zeta}} = GG^T$ and we define

$$T_x + T_{\tilde{x}} := (F + M \sum_{i=1}^{c} D_i \epsilon^i)\boldsymbol{x} + (\tilde{F} + \sum_{i=1}^{d} C_i \varepsilon^i \tilde{M})\tilde{\boldsymbol{x}}.$$

To derive $\hat{\Sigma}_{\breve{x}}$, we first regard the terms $\mathbb{E}_{\boldsymbol{x},\boldsymbol{\epsilon}} \left[ T_x T_x^T \right]$:

$$
\begin{aligned}
\mathbb{E}_{\boldsymbol{x},\boldsymbol{\epsilon}} \left[ T_x T_x^T \right] &= \mathbb{E}_{\boldsymbol{x},\boldsymbol{\epsilon}} \left[ (F + M \sum_{i=1}^{c} D_i \epsilon^i)\boldsymbol{x}\boldsymbol{x}^T (F^T + \sum_{i=1}^{c} \epsilon^i D_i^T M^T) \right] \\
&= \mathbb{E}_{\boldsymbol{\epsilon}} \left[ (F + M \sum_{i=1}^{c} D_i \epsilon^i) \, \mathbb{E}_{\boldsymbol{x}} \left[ \boldsymbol{x}\boldsymbol{x}^T \right] (F^T + \sum_{i=1}^{c} \epsilon^i D_i^T M^T) \right] \\
&= \mathbb{E}_{\boldsymbol{\epsilon}} \left[ (F + M \sum_{i=1}^{c} D_i \epsilon^i)\Upsilon_{xx}(F^T + \sum_{i=1}^{c} \epsilon^i D_i^T M^T) \right]
\end{aligned}
$$

$$= F\Upsilon_{xx}F^T + \mathbb{E}_{\epsilon}\left[F\Upsilon_{xx}\sum_{i=1}^{c}\epsilon^i D_i^T M^T\right] + \mathbb{E}_{\epsilon}\left[M(\sum_{i=1}^{c}D_i\epsilon^i)\Upsilon_{xx}F^T\right]$$

$$+ \mathbb{E}_{\epsilon}\left[M(\sum_{i=1}^{c}D_i\epsilon^i)\Upsilon_{xx}(\sum_{i=1}^{c}\epsilon^i D_i^T M^T)\right]$$

$$= F\Upsilon_{xx}F^T + F\Upsilon_{xx}\sum_{i=1}^{c}\mathbb{E}_{\epsilon}\left[\epsilon^i\right]D_i^T M^T + M(\sum_{i=1}^{c}D_i\mathbb{E}_{\epsilon}\left[\epsilon^i\right])\Upsilon_{xx}F^T$$

$$+ \mathbb{E}_{\epsilon}\left[M(\sum_{i=1}^{c}D_i\epsilon^i)\Upsilon_{xx}(\sum_{i=1}^{c}\epsilon^i D_i^T M^T)\right]$$

$$= F\Upsilon_{xx}F^T + \mathbb{E}_{\epsilon}\left[M(\sum_{i=1}^{c}D_i\epsilon^i)\Upsilon_{xx}(\sum_{i=1}^{c}\epsilon^i D_i^T M^T)\right]$$

$$= F\Upsilon_{xx}F^T + \mathbb{E}_{\epsilon}\left[M(\sum_{i=1}^{c}\sum_{j=1}^{c}D_i\epsilon^i\Upsilon_{xx}\epsilon^j D_j^T)M^T\right]$$

$$= F\Upsilon_{xx}F^T + \mathbb{E}_{\epsilon}\left[M(\sum_{i=1}^{c}D_i\epsilon^i\Upsilon_{xx}\epsilon^i D_i^T)M^T\right]$$

$$= F\Upsilon_{xx}F^T + M(\sum_{i=1}^{c}D_i\mathbb{E}_{\epsilon^i}\left[\epsilon^i\epsilon^i\right]\Upsilon_{xx}D_i^T)M^T$$

$$= F\Upsilon_{xx}F^T + M(\sum_{i=1}^{c}D_i\Upsilon_{xx}D_i^T)M^T,$$

where we defined the (raw) second moments of $x$ as

$$\Upsilon_{x} := \mathbb{E}_{x}\left[xx^T\right]$$
$$= \Sigma_{x} + \mu_x\mu_x^T.$$

With that, we obtain for $\mathbb{E}_{x,\epsilon}\left[T_xT_x^T\right] - \mu_x\mu_x^T$:

$$\mathbb{E}_{x,\epsilon}\left[T_xT_x^T\right] - \mu_x\mu_x^T = F\Upsilon_{xx}F^T + M(\sum_{i=1}^{c}D_i\Upsilon_{xx}D_i^T)M^T - F\mu_x\mu_x^T F^T$$

$$= F(\Upsilon_{xx} - \mu_x\mu_x^T)F^T + M(\sum_{i=1}^{c}D_i\Upsilon_{xx}D_i^T)M^T$$

$$= F\Sigma_{xx}F^T + M(\sum_{i=1}^{c}D_i\Upsilon_{xx}D_i^T)M^T$$

A similar derivation follows for $\mathbb{E}_{\tilde{x},\epsilon}\left[T_{\tilde{x}}T_{\tilde{x}}^T\right] - \mu_{\tilde{x}}\mu_{\tilde{x}}^T$, giving

$$\mathbb{E}_{\tilde{x},\epsilon}\left[T_{\tilde{x}}T_{\tilde{x}}^T\right] - \mu_{\tilde{x}}\mu_{\tilde{x}}^T = \tilde{F}\Sigma_{\tilde{x}\tilde{x}}\tilde{F}^T + \sum_{i=1}^{d}C_i\tilde{M}\Upsilon_{\tilde{x}\tilde{x}}\tilde{M}^T C_i^T.$$

Using these intermediate results, we can now compute $\hat{\Sigma}_{\breve{x}}$:

$$\hat{\Sigma}_{\breve{x}} = \mathbb{E}_{x,\epsilon,\tilde{x},\varepsilon}\left[(T_x + T_{\tilde{x}})(T_x + T_{\tilde{x}})^T\right] - \mathbb{E}_{x,\epsilon,\tilde{x},\varepsilon}\left[(T_x + T_{\tilde{x}})\right]\mathbb{E}_{x,\epsilon,\tilde{x},\varepsilon}\left[(T_x + T_{\tilde{x}})\right]^T$$
$$= \mathbb{E}_{x,\epsilon,\tilde{x},\varepsilon}\left[(T_xT_x^T + T_xT_{\tilde{x}}^T + T_{\tilde{x}}T_x^T + T_{\tilde{x}}T_{\tilde{x}}^T)\right] - (\mu_x + \mu_{\tilde{x}})(\mu_x + \mu_{\tilde{x}})^T$$
$$= \mathbb{E}_{x,\epsilon}\left[T_xT_x^T\right] - \mu_x\mu_x^T + \mathbb{E}_{x,\epsilon,\tilde{x},\varepsilon}\left[T_xT_{\tilde{x}}^T\right] - \mu_x\mu_{\tilde{x}}^T + \mathbb{E}_{x,\epsilon,\tilde{x},\varepsilon}\left[T_{\tilde{x}}T_x^T\right]$$
$$- \mu_{\tilde{x}}\mu_x^T + \mathbb{E}_{\tilde{x},\varepsilon}\left[T_{\tilde{x}}T_{\tilde{x}}^T\right] - \mu_{\tilde{x}}\mu_{\tilde{x}}^T$$

$$\overset{(*)}{=} F\Sigma_{xx}F^T + M(\sum_{i=1}^{c} D_i \Upsilon_{xx} D_i^T)M^T + \tilde{F}\Sigma_{\tilde{x}\tilde{x}}\tilde{F}^T + \sum_{i=1}^{d} C_i \tilde{M}\Upsilon_{\tilde{x}\tilde{x}}\tilde{M}^T C_i^T$$

$$+ \mathbb{E}_{\boldsymbol{x},\boldsymbol{\epsilon},\tilde{\boldsymbol{x}},\boldsymbol{\varepsilon}}\left[T_x T_{\tilde{x}}^T\right] - \boldsymbol{\mu}_x \boldsymbol{\mu}_{\tilde{x}}^T + \mathbb{E}_{\boldsymbol{x},\boldsymbol{\epsilon},\tilde{\boldsymbol{x}},\boldsymbol{\varepsilon}}\left[T_{\tilde{x}} T_x^T\right] - \boldsymbol{\mu}_{\tilde{x}} \boldsymbol{\mu}_x^T$$

$$= M(\sum_{i=1}^{c} D_i \Upsilon_{xx} D_i^T)M^T + \sum_{i=1}^{d} C_i \tilde{M}\Upsilon_{\tilde{x}\tilde{x}}\tilde{M}^T C_i^T + F\Sigma_{xx}F^T + \tilde{F}\Sigma_{\tilde{x}\tilde{x}}\tilde{F}^T$$

$$+ F\Sigma_{\boldsymbol{x}\tilde{\boldsymbol{x}}}\tilde{F}^T + \tilde{F}\Sigma_{\tilde{\boldsymbol{x}}\boldsymbol{x}}F^T$$

$$= M(\sum_{i=1}^{c} D_i \Upsilon_{xx} D_i^T)M^T + \sum_{i=1}^{d} C_i \tilde{M}\Upsilon_{\tilde{x}\tilde{x}}\tilde{M}^T C_i^T + \breve{F}\Sigma\breve{F}^T,$$

where in $(*)$ we used the previously derived results and defined $\breve{F}_t$ as $F_t$ and $\tilde{F}_t$ vertically stacked, i.e., $\breve{F}_t := \begin{bmatrix} F_t^T & \tilde{F}_t^T \end{bmatrix}^T$.

By putting everything together, we get $\boldsymbol{w}_{t+1} \sim \mathcal{N}(\hat{\boldsymbol{\mu}}_w, \hat{\Sigma}_w)$ with

$$\hat{\boldsymbol{\mu}}_w = F\boldsymbol{\mu}_x + \tilde{F}\boldsymbol{\mu}_{\tilde{x}} = \breve{F}\boldsymbol{\mu}_{\breve{x}},$$

$$\hat{\Sigma}_w = M(\sum_{i=1}^{c} D_i \Upsilon_{xx} D_i^T)M^T + \sum_{i=1}^{d} C_i \tilde{M}\Upsilon_{\tilde{x}\tilde{x}}\tilde{M}^T C_i^T + \breve{F}\Sigma\breve{F}^T + GG^T$$

$$= M(\sum_{i=1}^{c} D_i (\Sigma_{xx} + \boldsymbol{\mu}_x \boldsymbol{\mu}_x^T) D_i^T)M^T + \sum_{i=1}^{d} C_i \tilde{M}(\Sigma_{\tilde{x}\tilde{x}} + \boldsymbol{\mu}_{\tilde{x}} \boldsymbol{\mu}_{\tilde{x}}^T)\tilde{M}^T C_i^T + \breve{F}\Sigma\breve{F}^T + GG^T,$$

$$(18)$$

where $\breve{F}_t$ consists of $F_t$ and $\tilde{F}_t$ vertically stacked, i.e., $\breve{F}_t := \begin{bmatrix} F_t^T, & \tilde{F}_t^T \end{bmatrix}^T$.

### E.1 Partially-observable state

In case of partial observability, we have $\boldsymbol{w}_{t+1} = \begin{bmatrix} \boldsymbol{x}_{t+1}^T, \tilde{\boldsymbol{x}}_{t+1}^T, \boldsymbol{o}_{t+1}^T \end{bmatrix}^T$ with observations $\boldsymbol{o}_t$, so the goal becomes to approximate $p(\boldsymbol{x}_{t+1}, \tilde{\boldsymbol{x}}_{t+1}, \boldsymbol{o}_{t+1} \,|\, \boldsymbol{o}_{1:t})$. If $p(\boldsymbol{x}_t, \tilde{\boldsymbol{x}}_t \,|\, \boldsymbol{o}_{1:t})$ is distributed as in Eq. (17), Eq. (18) gives directly the formula for the approximation of $p(\boldsymbol{x}_{t+1}^T, \tilde{\boldsymbol{x}}_{t+1}^T, \boldsymbol{o}_{t+1}^T \,|\, \boldsymbol{o}_{1:t})$.

### E.2 Fully-observable state

In case of a fully-observable state, we have $\boldsymbol{w}_{t+1} = \begin{bmatrix} \boldsymbol{x}_{t+1}^T, & \tilde{\boldsymbol{x}}_{t+1}^T \end{bmatrix}^T$, with observation $\boldsymbol{o}_t = \boldsymbol{x}_t$, so we are interested in approximating $p(\boldsymbol{x}_{t+1}, \tilde{\boldsymbol{x}}_{t+1} \,|\, \boldsymbol{x}_{1:t})$. We assume $p(\tilde{\boldsymbol{x}}_t \,|\, \boldsymbol{x}_{1:t}) = \mathcal{N}\left(\boldsymbol{\mu}_{\tilde{x}|x}, \Sigma_{\tilde{x}|x}\right)$ and $\boldsymbol{x}_t$ to be observed and therefore deterministic. We can then informally write

$$p(\boldsymbol{x}_t, \tilde{\boldsymbol{x}}_t \,|\, \boldsymbol{x}_{1:t}) \sim \mathcal{N}\left(\begin{bmatrix} \boldsymbol{x}_t \\ \tilde{\boldsymbol{x}}_t \end{bmatrix} \,\middle|\, \boldsymbol{\mu}_t = \begin{bmatrix} \boldsymbol{x}_t \\ \boldsymbol{\mu}_{\tilde{x}|x} \end{bmatrix}, \Sigma_t = \begin{bmatrix} \mathbf{0} & \mathbf{0} \\ \mathbf{0} & \Sigma_{\tilde{x}|x} \end{bmatrix}\right).$$

Plugging this into Eq. (18), we obtain $\boldsymbol{w}_{t+1} \sim \mathcal{N}(\hat{\boldsymbol{\mu}}_w, \hat{\Sigma}_w)$, with

$$\hat{\boldsymbol{\mu}}_w = F\boldsymbol{\mu}_x + \tilde{F}\boldsymbol{\mu}_{\tilde{x}}$$

$$= F\boldsymbol{x}_t + \tilde{F}\boldsymbol{\mu}_{\tilde{x}|x},$$

$$\hat{\Sigma}_w = M(\sum_{i=1}^{c} D_i \Upsilon_{xx} D_i^T)M^T + \sum_{i=1}^{d} C_i \tilde{M}\Upsilon_{\tilde{x}\tilde{x}}\tilde{M}^T C_i^T + F\Sigma_{xx}F^T + \tilde{F}\Sigma_{\tilde{x}\tilde{x}}\tilde{F}^T$$

$$+ F\Sigma_{x\tilde{x}}\tilde{F}^T + \tilde{F}\Sigma_{\tilde{x}x}F^T + GG^T$$

$$= M(\sum_{i=1}^{c} D_i \boldsymbol{x}_t \boldsymbol{x}_t^T D_i^T)M^T + \sum_{i=1}^{d} C_i \tilde{M}\Upsilon_{\tilde{x}|x}\tilde{M}^T C_i^T + \tilde{F}\Sigma_{\tilde{x}|x}\tilde{F}^T + GG^T$$

$$= M(\sum_{i=1}^{c} D_i \boldsymbol{x}_t \boldsymbol{x}_t^T D_i^T)M^T + \sum_{i=1}^{d} C_i \tilde{M}(\Sigma_{\tilde{x}|x} + \boldsymbol{\mu}_{\tilde{x}|x} \boldsymbol{\mu}_{\tilde{x}|x}^T)\tilde{M}^T C_i^T + \tilde{F}\Sigma_{\tilde{x}|x}\tilde{F}^T + GG^T,$$

where the time-dependency of the matrices was omitted to enhance readability.

# F    Further information on applications

If not stated otherwise, we used 100 trajectories to determine the MLEs. For optimization, we ran the optimizer 10 times with random initial points and took the optimal value to avoid local optima.

## F.1    Reaching model

The reaching model was the same one used in the original publication [7], where all details can be found. The cost function which was minimized, is given by

$$\left(x_p(T) - x_p^*\right)^2 + (v \cdot x_v(T))^2 + (f \cdot x_f(T))^2 + \frac{r}{M-1} \sum_{k=0}^{M-1} u(k\Delta)^2,$$

where $x_p$ is the position, $x_v$ the velocity, $x_f$ the force, $x_p^*$ the target position, and $\Delta$ the time discretization, discretizing $T$ into $M$ time steps, i.e., $\Delta M = T$. Note that there is no explicit parameter for the cost of the end-point position needed because the other parameters are relative to this quantity.

We used the same model for the real reaching data, which were taken from the *Database for Reaching Experiments and Models*[4] and were previously published [55]. We took the horizontal component of reaching movements towards targets at 0 degrees (right of center) and truncated each trial so that it contained the movement only.

## F.2    Saccadic eye movement problem

We used the model by Crevecoeur and Kording [56] with a time discretization of 1.25 ms. The initial angle was set to $-10$ and the target angle to $10$ as shown in Figure 1b of the referenced paper.

## F.3    Random models

The random models were inspired by the work of Todorov [7]. For these models, the state space was four-dimensional with an additional dimension for modelling the target that the first dimension of the state should be controlled to. The action space was $n = 2$-dimensional. The matrices $A$, $B$ and $H$ of the dynamical system were randomly sampled with

$$A_{ij}, B_{ij}, H_{ij} \sim \mathcal{N}(0, 1).$$

$A$ and $B$ were normalized to 1 using the Frobenius norm. The additive noises $V$, $W$, and $E$ were sampled from LKJ-Cholesky distributions

$$V, W \sim \text{LKJ-Cholesky}(1)$$

and the multiplicative noises were sampled with

$$C_{ij}, D_{ij} \sim \text{Uniform}(0, 0.5).$$

The state cost matrices $Q_t$ were set to $Q_{1:T-1} = 0$ and $Q_T = \boldsymbol{d}\boldsymbol{d}^T$ with $\boldsymbol{d} = \begin{bmatrix} 1 & 0 & \dots & -1 \end{bmatrix}$ yielding a state cost at the last time step of $(\boldsymbol{x}_T - 1)(\boldsymbol{x}_T - 1)^T$. The control cost was parametrized with $R = \text{diag}[r_1, \dots, r_n]$. We used our maximum likelihood method to infer the parameters $r_1, \dots, r_n$.

# G    Additional results

In this section we will provide additional results for the reaching and random problems.

## G.1    Synthetic reaching data

### G.1.1    Comparison to a baseline

Fig. 6 shows the RMSEs of our method in comparison to the two baselines described in Section 4.1.

---

[4]https://crcns.org/data-sets/movements/dream/overview

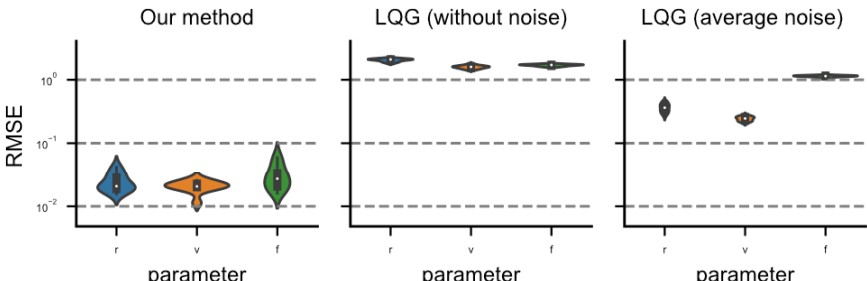

Figure 6: **Evaluation of the proposed inverse optimal control method in terms of RMSE of the maximum-likelihood parameter estimates.** Left: Our method, in which the likelihood is approximated via moment-matching. Middle: Standard LQG (without signal-dependent noise), for which the likelihood of the parameters given the data can be calculated in closed-form. Right: Standard LQG where the additive noise level was set to the average noise magnitude in the trajectories.

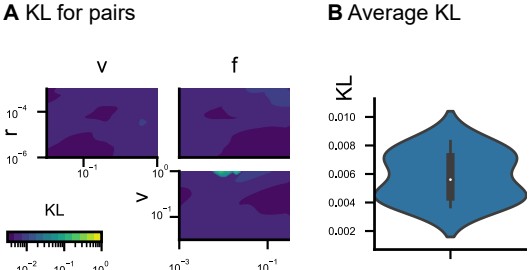

Figure 7: **KL divergences.** As a secondary evaluation metric, we show the KL divergence between trajectories simulated using the true parameters and trajectories simulated using the MLEs. The results are qualitatively similar to the RMSEs and show that the inferred parameters generate trajectories very similar to the observed data.

### G.1.2  Kullback–Leibler divergence as evaluation measure

As an alternative evaluation measure, we propose to compare the distributions induced by the true parameters and the maximum likelihood parameters. For this, we estimate empirical distributions of the trajectories by generating 10,000 trajectories and computing the mean and variance for each time step to approximate the distribution for each time step by a Gaussian. The Kullback–Leibler (KL) divergence between two Gaussian distributions $p = \mathcal{N}(\boldsymbol{\mu}_p, \Sigma_p)$ and $q = \mathcal{N}(\boldsymbol{\mu}_q, \Sigma_q)$ can be calculated in closed form as

$$D_{KL}(p \,\|\, q) = \frac{1}{2} \left[ \log \frac{|\Sigma_q|}{|\Sigma_p|} - k + (\boldsymbol{\mu}_p - \boldsymbol{\mu}_q)^T \Sigma_q^{-1} (\boldsymbol{\mu}_p - \boldsymbol{\mu}_q) + \mathrm{tr} \left\{ \Sigma_q^{-1} \Sigma_p \right\} \right].$$

Instead of using the KL divergence directly, which is not symmetric, we consider instead a symmetric version,

$$\frac{1}{2} D_{KL}(p \,\|\, q) + \frac{1}{2} D_{KL}(q \,\|\, p),$$

and compute the mean over time to aggregate the values over time.

The results like in Fig. 2 E and F with KL divergence instead of RMSE as a metric are shown in Fig. 7.

### G.1.3  Partially observable state

We evaluate a version of the reaching model in which the experimenter observes only the position and treats velocity and acceleration as latent variables. The results are qualitatively similar to the fully observed case. However, there are regions in the parameter space where estimates are worse (Fig. 8 A). Additionally, estimates of the parameters representing the penalty on velocity ($v$) and acceleration ($f$) are worse by an order of magnitude compared to the fully observed case (Fig. 8 B),

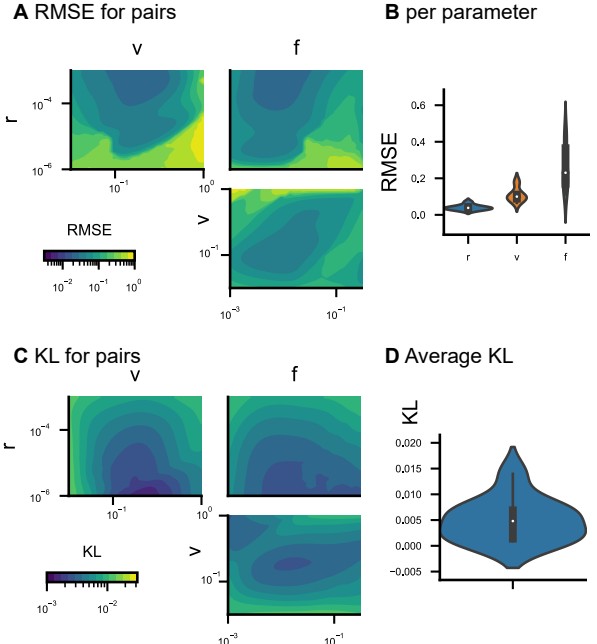

Figure 8: **Evaluation of partially observed model.** RMSEs and KLs for partially observed reaching model.

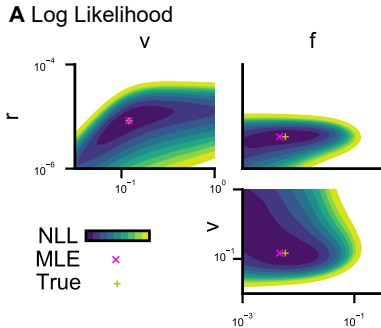

Figure 9: **Negative log likelihod of the partially observed model.**

which is to be expected when only the position is observed. Specifically, the average RMSEs are $4.0 \times 10^{-2}$ ($r$), $1.1 \times 10^{-1}$ ($v$), and $2.6 \times 10^{-1}$ ($f$). However, the parameter errors do not result in large differences in the KL divergence of the simulated trajectories (position only) w.r.t. the observed trajectories, so we suspect that the higher RMSEs in estimated parameters are due to ambiguities in the trajectories for this particular model.

### G.1.4 Evaluation of moment matching approximation

In Section 3.2 we introduced a moment matching approximation to make computation of the likelihood tractable. An experimentalist comparing an optimal control model to experimental data might be interested in the influence of this approximation on trajectories. For the reaching model from Section 4.1, we therefore compare the empirical distributions over trajectories estimated using Monte Carlo rollouts (using 10,000 trajectory samples) to the approximate distribution over trajectories determined using our method (given the true parameters). The difference in symmetrized KL (see Appendix G.1.2) between the empirically estimated distribution and our approximation is found to be $1.60 \times 10^{-3}$. Additionally, we compare this result to a baseline by replacing the signal-dependent noise by additive noise, for which the trajectory distribution can be calculated in closed form. The additive noise magnitude is chosen as the average of the signal-dependent noise magnitudes for the whole trajectories. Note that this quantity is not directly available and therefore has to be also

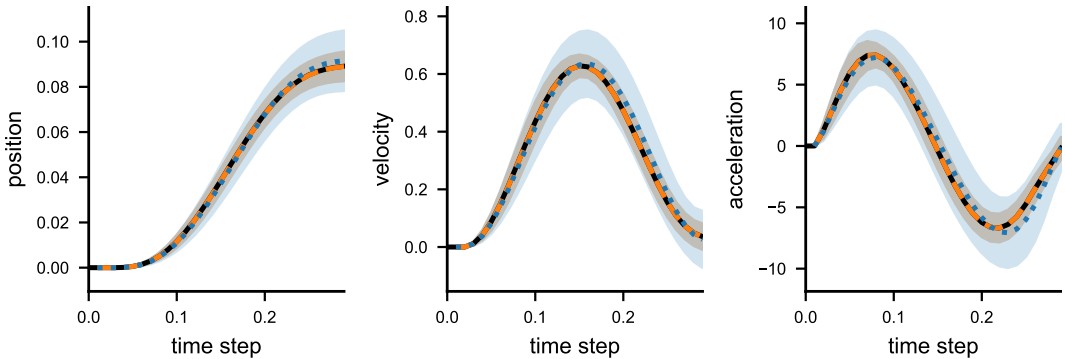

Figure 10: **Empirical evaluation of the moment matching approximation.** Trajectory distributions of the reaching task using a Monte-Carlo approximation (mean: black solid line, $2 \times$ STD in gray), our moment-matching approximation (orange, dashed), and a baseline where the signal-dependent noise is replaced by additive noise (blue, dotted). The gray and orange areas are hardly distinguishable visually, showing that the moment matching approximation estimates the trajectory distribution very precisely. The baseline overestimates the variance through the signal-dependent noise in early time steps, leading to an overall too high variance of the trajectories.

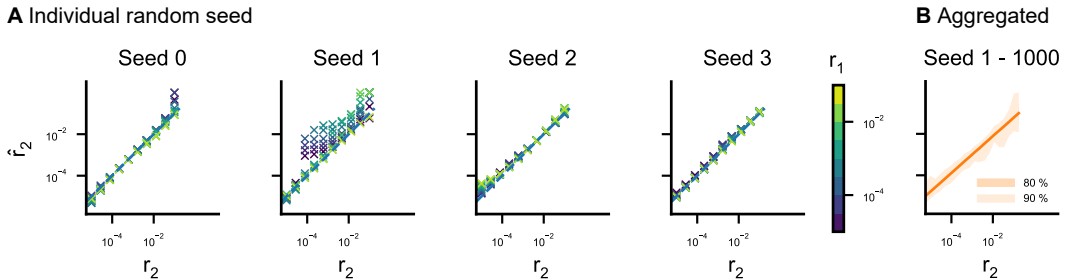

Figure 11: **Random problems (second parameter).** **A** MLEs for a range of parameter values of $r_2$ for different random problems and different values of $r_1$. **B** Aggregated results (median, percentiles) for 1000 random problems.

estimated, e.g., via Monte-Carlo rollouts. The difference in symmetrized KL between the empirical estimate and the baseline is $6.05$. A plot of the resulting distributions is shown in Fig. 10. As expected, the moment matching approximation estimates the trajectory distribution very precisely in comparison to the baseline.

## G.2 Random problems

In Fig. 11 A we show the errors for a range of parameter values $r_2$ for different random models. The median and quantiles for the results of 1000 random problems are shown in Fig. 11 B. One can observe that the estimates are generally very close to the true parameters.