# OpenReview forum: "Inverse Optimal Control Adapted to the Noise Characteristics of the Human Sensorimotor System"
_NeurIPS.cc/2021/Conference — NeurIPS 2021 Poster_

### Official Review · Reviewer_9TgB · 2021-07-12

**Rating:** 6
**Confidence:** 3

**Summary:**

This paper formulates an inverse optimal control solution for a control system with partial observability and noise properties characteristic of the human sensorimotor system - that is, control signal-dependent motor noise, and state-dependent observation noise. The contribution is that this formulation can enable an experimenter to estimate parameters of the cost function of a control system (e.g. a human’s motor control system) under assumptions that that control system acts optimally, has full knowledge of the system dynamics (an accurate internal model), and has a functional form similar to that of the proposed formulation (a variant of the LQG case). Through a series of experiments comparing their inverse model to simulated reaching data, real macaque reaching data, simulated saccadic eye movements, and simple non-biological control problems, the authors show that their method can accurately estimate unknown cost function parameters to reverse-engineer an approximate model of the control system in question.

**Limitations And Societal Impact:**

Yes.

**Main Review:**

The paper takes a previous formulation by Todorov (2007) of the control problem faced by an agent with noise properties similar to that of the human sensorimotor system (and who acts optimally under full knowledge of its own dynamical system), and instead considers the inverted problem of how to discover the cost function parameters of this agent. The authors use probabilistic methods to derive this formulation and provide an algorithm for computing the approximate likelihood of the control parameters. The main contribution is thus to combine inverse optimal control in the LQG setting with the signal-dependent sensory and motor noise modelled by Todorov.
The authors validate that their method works in a series of well conceived experiments using primate reaching and saccade movement simulators, real primate psychophysical reaching data (previously published) as well as a classic control problem. The method does seem to work well, based on the data presented.
The surrounding literature seems well reviewed, and I agree that this method could in theory be applied to estimating control properties in psychophysical motor experiments, if of course, the objective is still to estimate what trade-offs a human or non-human primate is making when they move a single part of their body in a highly constrained environment (under the assumption of optimality, and that the agent controls using the same state-space basis as the model assumes). Psychology and neuroscience experiments moving towards more naturalistic movements and environments will pose a major problem for this form of control work, as the state-space grows and becomes more difficult to explicitly specify.
The point of modelling a separation between the agent’s beliefs and those of the experimenter was a little lost on me, as it’s not clear to me what exact scientific application this could have.

*EDIT:*
I thank the authors for clarifying their motivation for modelling a separation between the agent's and experimenter's beliefs, and on reflection I agree this contribution is a valuable one. I have increased my overall score accordingly, and would recommend that the authors attempt to make the points they have listed here in their response clearer in the paper if possible. Thanks also for the clear response to my note about the limitations of these traditional frameworks for assessing motor behaviour. I agree that acknowledging this point in the limitations of the paper would be helpful for readers.

**Time Spent Reviewing:**

2.5

---

> ### Author Response · Authors · 2021-08-10
> **Reply to reviewer**
>
> We like to thank the reviewer for commending on the “well conceived experiments”.
>
> *** I agree that this method could in theory be applied to estimating control properties in psychophysical motor experiments, if of course, the objective is still to estimate what trade-offs a human or non-human primate is making when they move a single part of their body in a highly constrained environment (under the assumption of optimality, and that the agent controls using the same state-space basis as the model assumes). Psychology and neuroscience experiments moving towards more naturalistic movements and environments will pose a major problem for this form of control work, as the state-space grows and becomes more difficult to explicitly specify. ***
>
> We fully agree with the reviewer that psychology and neuroscience are again in a phase of moving towards more “naturalistic” tasks. Without spoiling our identity, we actually have been contributing to this trend and we fully support this trend. This is one of the prime motivations for the development of inverse optimal control algorithms as in the current manuscript: In more naturalistic tasks it is quite difficult to guess the cost function a priori. The current “forward” optimal control formulations have led researchers to having to guess a cost function and then analyzing the behavioral data under the assumption of optimal behavior. Instead, our approach enables recovering the implicit costs and benefits in sensorimotor behavior, including trade-offs resulting e.g. from biomechanical or behavioral or cognitive costs. Thus, in our view, inverse optimal control methods are an important avenue to better understand more naturalistic behavior.
>
> On the other hand, moving to more naturalistic movements does not nullify what has been or can be learned experimentally and theoretically about human and animal sensorimotor behavior through paradigms involving short duration movements. As an example, Krakauer, Hadjiosif, Xu, Wong, Haith (2019) explicitly discuss the dichotomy of single, short movements and planning of extended, naturalistic action sequences. This relates to the large body of work that has conceptualized more naturalistic behavioral sequences as being composed of elemental sensorimotor acts, for which much neurophysiological support exists. Thus, the relevance of short, individual movements such as a single saccade or a grasping movement, which have been studied in isolation, to the understanding of more naturalistic behavior is an empirical question that has not been answered conclusively and we foresee our proposed method to contribute to answering this question.
>
> Finally, state space models describing limb kinematics or muscle dynamics are derived by the physical relationships between time, space, mass, forces, and energy. There is quite some work showing that these state space models do agree well with empirical data ranging from quantifying fatigue to calorimetric measurement of energy expenditure. These experiments suggest that the nervous system does have an internal model of its body’s dynamics that is quite accurate.
>
> Taken together, we think that the proposed computational framework and method will be of great utility to researchers in movement science, neuroscience, and cognitive science to uncover the information processing of biological systems in terms of a computational level description and help moving in the direction of more naturalistic movements. We will discuss this point in the limitations section of the final manuscript.
>
> *** The point of modelling a separation between the agent’s beliefs and those of the experimenter was a little lost on me, as it’s not clear to me what exact scientific application this could have. ***
>
> The separation between the agent’s and the experimenter’s belief has also been motivated by the move towards more naturalistic settings. In an experiment, we as researchers might only have noisy observations of the true trajectories (e.g. from a motion tracking system) and some dimensions of the state and the actions might be not observable at all. As a straightforward example, a movement’s acceleration might be unobserved experimentally but computed by some discrete approximation to differentiation on sampled positional data, with the common problem of having to balance amplifying measurement noise and smoothing data. But the experimental subject receives different noisy measurements of the relevant variables via their sensory system and computes its own belief over these variables. By distinguishing between the agent’s belief and that of the experimenter, which might have different levels of uncertainty, the analysis moves beyond claiming optimality from ideal observer models and putative cost functions but actually measures subjective uncertainties and costs of the subject. See e.g. the following reference for a similar rationale:
>
> M. Kwon, S. Daptardar, P.R. Schrater, Z. Pitkow. Inverse Rational Control with Partially Observable Continuous Nonlinear Dynamics. Advances in Neural Information Processing Systems, 2020. In our view, this is an important step in the direction of studying more naturalistic movements.
>
> Accordingly, we agree that traditional ideal observer analysis, in which the experimenter models behavior under the assumption that the subject has full knowledge about the task goal, the stimulus dynamics, is limited. This is precisely the reason for moving to an inverse optimal control analysis of experimental data, in which we allow for the possibility that subjects have subjective costs and benefits, which guide their behavior. This has the potential of quantifying biomechanical or cognitive costs unknown to the researcher. Thus, again, explicitly modeling the task from the subject’s perspective and distinguishing this from the perspective of the researcher is precisely motivated by abandoning ideal observer analysis with optimality claims based on putative cost functions.
>
> Furthermore, in our probabilistic model, we also infer the agent’s belief from the researcher’s perspective (see Fig. 2C), a quantity that is usually difficult to determine but of high interest for interpreting behavior. For example, it can be used for distinguishing between which biases and what part of variability in the subjects’ behavior might be due to perceptual influences and which might be due to costs or action variability. Thus, in our view, our approach enables recovering the costs and benefits implicit in human sequential sensorimotor behavior, thereby reconciling normative and descriptive approaches in a computational framework. Overall, in our view, this contributes to the understanding of neural information processing systems in their biological, technological, mathematical, and theoretical aspects. We hope that this clarifies why we think that  modelling the agent’s beliefs and those of the experimenter separately is important.

---

### Official Review · Reviewer_QXkU · 2021-07-12

**Rating:** 7
**Confidence:** 3

**Summary:**

This paper presents an algorithm to infer cost parameters of LQG problems given trajectories. The effectiveness of the algorithm is demonstrated on simulated arm and eye movements as well as random control problems.

**Limitations And Societal Impact:**

They are well discussed.

**Main Review:**

Pro:

1. The proposed method is demonstrated on a wide range of tasks, including real world dataset of a rhesus monkey reaching for objects.

2. The proposed method is able to infer the parameters under partially observed trajectories.

3. Latent velocity and acceleration can also be estimated using the proposed approach.

Con:

1. Only linear dynamics and quadratic const are considered. Although the limitation is discussed.

Questions:

1. It is not clear to me if the dynamics is assumed to be known (i.e, A, B matrix). If it is known, then how is the dynamics for the monkey estimated?

2. How important is the assumption that A, B is constant? Can we work with a time varying A, B in the current framework. This will be important to extend this to iLQG settings.

3. An important application of inferring the cost parameters is using it to predict behaviors that is not in the dataset or generate controllers that can mimic the behaviors of the agent. Any thoughts on this?

**Time Spent Reviewing:**

3

---

> ### Author Response · Authors · 2021-08-10
> **Reply to reviewer**
>
> We like to thank the reviewer for the positive evaluation of our manuscript.
>
> *** It is not clear to me if the dynamics is assumed to be known (i.e, A, B matrix). If it is known, then how is the dynamics for the monkey estimated? ***
>
> We are sorry for not having been clearer about the problem setting, particularly which matrices are given and what quantities are inferred. In general, our probabilistic formulation of the inverse optimal control problem allows inferring parameters of any of the matrices of the system by evaluating the likelihood function w.r.t. those parameters. For the monkey data example, we used the same reaching model as in Section 4.1, because it widely applies to many reaching applications. Sensorimotor neuroscientists often have quite accurate linear models for the tasks they are studying and have spent decades developing useful approximate linear models of the kinematic chains in the body and even muscle dynamics. Motivated by the fact that the costs and benefits, i.e. the cost function, is usually the least understood quantity in sensorimotor research and of particular interest when investigating optimality of behavior, we have treated all matrices except $Q$ and $R$ as given in our examples in this paper. We thus constrained the problem to inferring the subjective costs given the rest of the system, which is a reasonable assumption for a behavioral experiment, in which the experimenter has a good model of the task and the subjects’ noise characteristics. However, by parameterizing the matrices $A$ and $B$, one could in principle also estimate a model from the observed data given our probabilistic formulation. We will include a clear statement of this in the final version of the manuscript. We will also reference previous work that has focused on estimating individual components of this model through experiments, e.g. reference [22].
>
> *** How important is the assumption that A, B is constant? Can we work with a time varying A, B in the current framework. This will be important to extend this to iLQG settings. ***
>
> Yes, thank you for asking about the importance of the assumption of stationarity of the matrices A and B. In our inference algorithm, all matrices can in principle be time-varying. One would simply need to include the time-varying $A_t$ and $B_t$ in the definitions of the matrices of the joint dynamical system defined in Section 3.1, which are already time-varying due to the time-varying $K_t$ and $L_t$. For the sake of notational simplicity and to keep the examples straightforward, we used time-invariant matrices $A$ and $B$. The time-varying versions will mainly play a role for an iLQG extension, which is left as future work. We will mention this fact in the final version of the manuscript.
>
> *** An important application of inferring the cost parameters is using it to predict behaviors that is not in the dataset or generate controllers that can mimic the behaviors of the agent. Any thoughts on this? ***
>
> Yes, thank you for pointing out that we should include both a discussion of transfer learning of learnt cost functions to new settings as well as potential applications in the area of imitation and apprenticeship learning. Both are very good points, which should be addressed in our conclusion section by discussing possible applications of our method. We will include an additional paragraph in the discussion section and cite work on apprenticeship learning:
>
> P. Abbeel & A.Y. Ng. Apprenticeship learning via inverse reinforcement learning. In Proceedings of the twenty-first international conference on Machine learning, 2004.
>
> J. Ho, S. Ermon. Generative adversarial imitation learning. In Advances in Neural Information Processing Systems, 2016.
>
> T. Osa, J. Pajarinen, G. Neumann, J. Bagnell, P. Abbeel, & J. Peters. An Algorithmic Perspective on Imitation Learning. Foundations and Trends in Robotics, 7(1-2), 1-179, 2018
>
> and on transfer learning within RL:
>
> M. E. Taylor, P. Stone. Transfer learning for reinforcement learning domains: A survey. Journal of Machine Learning Research 10.7, 2009.
>
> A. Lazaric. Transfer in reinforcement learning: a framework and a survey. Reinforcement Learning. Springer, Berlin, Heidelberg, 143-173, 2012.
>
> Z. Zhu, K. Lin, J. Zhou. Transfer learning in deep reinforcement learning: A survey. arXiv preprint arXiv:2009.07888, 2020.
>
> Unfortunately, current experimental paradigms in movement science, neuroscience, and cognitive science, have not yet widely adopted experiments to address these types of questions systematically. Thus, transfer is often limited to available data on very similar experimental settings. The generation of controllers that can mimic the behaviors of the agent is an interesting application that we will discuss.
> If the reviewer would like us to cite other specific works in this area that are additionally relevant, we are happy to do so.

---

### Official Review · Reviewer_ueSj · 2021-07-17

**Rating:** 8
**Confidence:** 4

**Summary:**

This paper presents a theoretical framework for cost function estimation for linear, quadratic, Gaussian systems with action- and state-dependent noises.
Algorithms for estimating the probabilities of latent variables and observed signals are derived for both the cases of complete and partial observation of the task variables by the experimenter. Then a gradient-free optimizer BOYAQA is used to obtain the maximum likelihood estimate of the parameters of the objective function.
The performance was examined with 1D reaching simulation and real arm reaching and saccadic eye movement experimental data.


**Limitations And Societal Impact:**

These are well considered and documented.

**Main Review:**

Originality:
The method is straight forward applications of dynamic Bayesian inference and maximum likelihood estimate, but the formulation of inference by the agent and by the experimenter as graphical models as presented in Figure 1 is eye-opening.

Quality:
The mathematical derivation is solid and clear.
In the 1D reaching simulation, why isn't the end-point position error included in the cost function? Is it supposed to be zero cost as long as the end point is within a target range?
The case of center-out arm reaching clearly violates the linear assumption. The method seems to be working nevertheless, and the reason why would be worth discussing.

Clarity:
Probably not all the readers are familiar with the impact of Harris & Wolpert paper, it may worth more intuitive explanation.
If I understand correctly, the method assumes that the experimenter knows the dynamics model (A, B, V, Ci, H, W, Di). It is worth mentioning how they can be estimated

Significance:
Although the theory was derived under linear, quadratic, Gaussian assumptions, its applicability with real arm and eye movement data is encouraging.


**Time Spent Reviewing:**

2h

---

> ### Author Response · Authors · 2021-08-10
> **Reply to reviewer**
>
> We like to thank the reviewer for the positive evaluation of our manuscript and calling our formulation of the problem setting “eye-opening”.
>
> *** In the 1D reaching simulation, why isn't the end-point position error included in the cost function? Is it supposed to be zero cost as long as the end point is within a target range? ***
>
> We are sorry for not explaining more clearly how the end-point position error is included in the cost function. The endpoint error enters the cost function because the $Q$ matrix at the final time step contains entries which result in a quadratic cost term ${(p - p^*)}^2$ for the distance between hand position $p$ and target position $p^*$ when evaluating $x’Qx$. There is no separate parameter for this term in the cost function needed, because the other terms (velocity and acceleration end-point costs, action costs) are relative to this position end-point error (see Section 6.1 of Todorov, 2005).
>
> *** The case of center-out arm reaching clearly violates the linear assumption. The method seems to be working nevertheless, and the reason why would be worth discussing. ***
>
> Sorry for not being more clear on the details of the used model. From Todorov (2005, Section 6.1):
>
> “We model a single-joint movement (such as flexing the elbow) that brings the hand to a specified target. For simplicity, the rotational motion is replaced with translational motion; the hand is modeled as a point mass ($m = 1$ kg) whose one-dimensional position at time $t$ is $p(t)$. The combined action of all muscles is represented with the force $f(t)$ acting on the hand. The control signal $u(t)$ is transformed into force $f(t)$ by adding control-dependent multiplicative noise and applying a second-order muscle-like low-pass filter (Winter, 1990) of the form $\tau_1 \tau_2 \ddot{f}(t) + (\tau_1 + \tau_2) \dot{f}(t) + f(t) = u(t)$, with time constants $\tau_1 = \tau_2 = 0.04$ sec. Note that a second-order filter can be written as a pair of coupled first-order filters (with outputs $g$ and $f$) as follows: $\tau_1 \dot{g}(t) + g(t) = u(t), \tau_2 \dot{f}(t) + f(t) = g(t)$.”
>
> Thus, in this case the end-effector in the reaching and tracking tasks is approximated with a linear system under some simplifying conditions, resulting in purely translational motion of the hand emulating experimentally observed linear trajectories and combining the action of all muscles into one force acting on the end-effector. The resulting position trajectories are then a non-linear function of position as the state variable includes velocity and acceleration.
>
> *** Probably not all the readers are familiar with the impact of Harris & Wolpert paper, it may worth more intuitive explanation. ***
>
> Yes, thanks! Harris & Wolpert (1998) made a seminal contribution to sensorimotor neuroscience by unifying several empirical phenomena, including smooth bell-shaped velocity profiles and Fitts’ law, based on the theoretical assumption that variability in control signals is signal-dependent. It also shifted the explanation of movements as arising from minimization of functionals describing movement costs to a description involving the actual goal of a movement task. We will emphasize this point both in the second paragraph of the introduction and in the conclusion section of the final version of the manuscript.
>
> *** If I understand correctly, the method assumes that the experimenter knows the dynamics model (A, B, V, Ci, H, W, Di). It is worth mentioning how they can be estimated ***
>
> Yes, thank you for pointing out that we should more clearly mention how research in sensorimotor control has theoretically modeled and empirically measured the model’s other quantities. As mentioned to reviewer 1 (VUPJ), our probabilistic formulation of the inverse optimal control problem allows inferring parameters of any of the matrices of the system by evaluating the likelihood function w.r.t. those parameters. Motivated by the fact that the costs and benefits, i.e. the cost function is usually the least understood quantity in sensorimotor research and of particular interest when investigating optimality of behavior, we have treated all matrices except Q and R as given in our examples in this paper. We thus constrained the problem to inferring the subjective costs given the rest of the system, which is a reasonable assumption for a behavioral experiment, in which the experimenter has a good model of the task and the subjects’ noise characteristics. We will include a clear statement of this in the final version of the manuscript. We will also reference previous work that has focused on estimating individual components of this model, e.g. reference [22].

---

### Official Review · Reviewer_VUPJ · 2021-07-23

**Rating:** 4
**Confidence:** 4

**Summary:**

The authors present a method for recovering cost functions from observed state trajectories through inverse optimal control with signal-dependent noise for linear systems and quadratic costs.
They formulate a POMDP and distinguish between the agent’s and the experimenter's inference problems since the agent does not have access to the experimenter's hidden states.
They derive the joint dynamics of the state and estimate by stacking the individual transition functions.
The resulting probabilistic belief transition function for the linear-quadratic Gaussian problem with signal-dependent noise is approximated through moment matching since it involves a product of two Gaussians (which does not result in a Gaussian).
In a similar fashion to stacking the agent and experimenter dynamics, the authors suggest additionally stacking the observer dynamics and solving the transition jointly for the case that they don’t have access to the full state.
The approach is validated on synthetic and experimental data and applied to human sequential sensorimotor behavior.


**Limitations And Societal Impact:**

Yes

**Main Review:**

Strengths:
- The problem is interesting and the solution is appropriate for a limited number of use cases.
- A concise background and description of LQG. Although this is nearly 1:1 from [1] I believe it is necessary to follow the notation and problem setting.
- Visualizations are intuitive and easy to understand
- Focus on methods that increase interpretability of behavior
- Good discussion of limitations in the last section

Weaknesses:
- It would have been good to more clearly state which matrices are given and what quantities are inferred.
Introducing an algorithm before section 3.1 would be helpful to make this more clear.
- Similarly, it would have been good to detail more closely how K_t and L_t are iteratively determined.
- I believe a more complete discussion about the impact of the moment matching assumption is needed. This could be done either theoretically or empirically through Monte-Carlo simulation and discussion of an intuitive example.
- The related work section does a good job referring to previous work, but overall I feel like it is too dated. There have been many more recent works in this domain over the last 5 years.
- “While more recent and more general methods for optimal control in high-dimensional continuous domains exist, they rely on function approximation methods, which is useful in engineering applications but may not provide a computational level explanation of behavior” I feel that this argument needs to be substantiated since similar approaches in the continuous inverse optimal control setting exist which could potentially be adapted to the signal-dependent noise setting [2]. These do not necessarily rely on NN function approximation, allow for interpretable non-linear features, and are computationally tractable.
- Overall the applicability of the approach is very limited. While the authors hint at a more capable integration with iLQG, I had hoped to see this in this paper already.
- The evaluation is nicely visualized but more than 1-dimensional experiments are needed. While the authors state that the approach does not scale to higher dimensions, I would have liked to see a dimensionality of at least 3.
- The evaluation lacks comparison to any baselines. No ablations are done. Additionally to related work, it would be good to ablate how well the costs be inferred through more primitive versions of the algorithm (e.g. no noise assumption, no signal-dependent noise, etc.)

Overall, I believe the work is interesting and timely but needs improvements, focusing on the evaluation. I would also like to encourage the authors to increase the scope and applicability of the work.

Typos:
“an which in general is intractable”
“requires a progressively larger number function evaluations”

[1] Todorov, Emanuel. "Stochastic optimal control and estimation methods adapted to the noise characteristics of the sensorimotor system." Neural computation 17.5 (2005): 1084-1108.

[2] Levine, Sergey, and Vladlen Koltun. "Continuous inverse optimal control with locally optimal examples." arXiv preprint arXiv:1206.4617 (2012).

EDIT: I thank the authors for addressing my concerns with respect to the impact of the moment matching approximation and incorporating two baselines. Nonetheless, I believe that using a single experiment is not sufficient for this conference. Due to the limited experiments and limited applicability of the method (without a simple iLQG integration, which should not be a huge effort), I believe this paper needs additional work and would like to keep the current rating as is.

**Time Spent Reviewing:**

6

---

> ### Author Response · Authors · 2021-08-10
> **Reply to reviewer**
>
> We like to thank the reviewer for the comments and specific request, which we have hopefully addressed satisfactorily.
>
> We are sorry for not having been more explicit about the problem setting, particularly which matrices are given and what quantities are inferred. In general, our probabilistic formulation of the inverse optimal control problem allows inferring parameters of any of the matrices of the system by evaluating the likelihood function w.r.t. those parameters. In the current manuscript, we have derived the inference algorithm for the Q and R matrices assuming knowledge of the other quantities in our examples. Thus, we constrained the problem to inferring the subjective costs given the system. The reason for this choice is motivated by the literature on behavioral experiments, in which the experimenter has a good model of the task and the subjects’ noise characteristics, but may not know the implicit behavioral goal, e.g. the benefits of task achievement and the biomechanical or cognitive costs of movements. We will include a clear statement of this and add an algorithm box in the final version of the manuscript.
>
> We are sorry for the confusion regarding the iterative computation of K_t and L_t and will add the equations from Todorov (2005), which describe how K_t and L_t are iteratively determined, to our supplementary material.
>
> Thank you for pointing out that a more thorough discussion of the impact of the moment matching approximation would be beneficial. The validation on synthetic reaching data shows that the true parameters can be reliably recovered using our approximate method, therefore the moment matching approximation seems to have only minor impact, at least for the considered problems. It is a good recommendation, though, to analyse the errors introduced by the moment matching in isolation and to add it to our paper. We thank the reviewer for this hint.
> An experimentalist comparing an optimal control model to experimental data might be interested in the influence of this approximation on trajectories. For this we will additionally report (1) the empirical distributions over trajectories estimated using Monte Carlo rollouts and comparisons to the approximate distribution over trajectories obtained by our method involving moment matching. Additionally, we will include (2) comparison to a baseline by replacing the signal-dependent noise by additive noise for which the trajectory distribution can be calculated in closed form. As expected, our approximation is very close to the distribution obtained by Monte Carlo simulation and the baselines deviate strongly. Specifically, the difference in symmetrized KL (see Appendix F.1.1) between the empirically estimated distribution and our approximation is 1.60 * 10^-3 for (1), while the symmetrized KL between the empirical estimate and the baseline (2) is 6.05. We will add these results and a plot visualizing these differences to the final version of the manuscript in the supplementary material.
>
> We thank the reviewer for suggesting to include more related work. We are a bit unclear about the intended “domain”, though. In our related work section, we have on the one hand focused on inverse optimal control approaches in the domain of the behavioral sciences, where references include papers published in 2020 and 2021. On the other hand, we have reviewed related work in the LQR / LQG domain, which includes references published in 2019. We are aware of the fact that there is a vast array of recent work on control in high dimensional spaces and on the general inverse optimal control problem. But, we are not aware of work in these areas that addresses the noise characteristics of the human sensorimotor system. Nevertheless, to address this point, we have decided to cite the following papers on recent approaches to optimal control:
>
> Levine, S., & Koltun, V.. Guided policy search. In International conference on machine learning, 2013.
>
> Levine, S., Finn, C., Darrell, T., & Abbeel, P.. End-to-end training of deep visuomotor policies. The Journal of Machine Learning Research, 17(1), 1334-1373, 2016.
>
> Additionally, based on the request by a different reviewer, we included references to transfer learning in RL and apprenticeship learning, which include more recent references. Additional references to recent work in inverse optimal control are included in response to the following point by the reviewer.
> If the reviewer would like us to cite other specific works in this area, we are happy to do so.
>
> We apologize for not being precise enough in our statement about function approximation so that it could be easily misunderstood.
> Many of the state of the art IRL methods make use of NN function approximators, e.g. those involving GANs. Those methods that use a NN to approximate the reward function make it hard to interpret it, a fact that has been repeatedly noted. These methods approximating a cost function through NNs were meant in our statement.
> We agree that there are other methods with interpretable non-linear features such as the one suggested by the reviewer. We also agree that these could in principle be used to create more ‘general’ methods, referring to relaxing LQG assumptions. However, there is still a lot of work to do to achieve this. The work recommended by the reviewer, as an example, considers deterministic MDPs (even without additive noise) and assumes full-observability. Another difficulty is that this method, like most other IRL methods, models expert demonstrations as Boltzmann distribution and is agnostic about where the noise in the trajectories comes from. The present model, on the other hand, is based on an accurate model about the noise, which has been established by a considerable body of research.
> In the final version, we will reword the passage and include a hint to the suggested citation as a suitable starting point for future work making use of interpretable non-linear features.
> Accordingly, we now also cite the following paper, as suggested by the reviewer:
>
> Levine, Sergey, and Vladlen Koltun. "Continuous inverse optimal control with locally optimal examples." In Proceedings of the International Conference on Machine Learning, 2012
>
> Additionally, to address this point by the reviewer, we have decided to cite the following papers referring to more recent work on inverse optimal control and IRL:
>
> J. Fu, K. Luo, and S. Levine. Learning robust rewards with adversarial inverse reinforcement learning. In International Conference on Learning Representations, 2018.
>
> A. J. Chan & M. van der Schaar. Scalable Bayesian Inverse Reinforcement Learning. In International Conference on Learning Representations, 2020.
>
> Again, if the reviewer would like us to cite other specific works in this area, we are happy to do so.
> We hope that this clarifies and substantiates our point and helps improve the manuscript.
>
> We agree with the reviewer that there is always room for extensions and further research in every paper. In our view, the present manuscript makes significant progress in that it derives the probabilistic inference process for estimating the cost matrices R and Q for the optimal control problem with linear dynamics, quadratic costs, and signal dependent noise, which has been used to gain invaluable insight into animal and human sensorimotor behavior. We extend this setting to the partially observable case by allowing state variables to be hidden. This will allow researchers in the behavioral sciences, neuroscience, and computational cognitive science to infer the costs and benefits implicit in sensorimotor behavior measured in experiments thereby allowing a better understanding of the brain’s control of sensorimotor behavior. Thus, our view is that the approach has broad applicability at least in the study of information processing systems in biological systems. We agree that further research can address the iLQG setting based on the present work as we have described in the discussion section.
>
> Sorry for not making it more clear that the 1-dimensional visualizations pertain to higher dimensional experiments. Each plot shows individual one dimensional trajectories of single state variables over time stemming from a higher dimensional state vector. so, in our reaching example, which we have taken from Todorov (2005), the state space is 5-dimensional and the parameter space is 3-dimensional. We will point out more clearly in the main text that the experiments pertain to state spaces that have more than one dimension.
>
> Thank you for your suggestion to include more comparisons, especially a baseline. In the final version we will report the results of the following two baselines.
> A first baseline is obtained by running a version of the algorithm without signal-dependent noise (i.e. using the basic LQG during inference) and the results on the reaching problem in terms of the RMSE are worse by roughly two orders of magnitude (RMSE of 1.766 vs 0.027). This means that the trajectories sampled using the parameters inferred without signal-dependent noise are significantly different from the actual trajectories (cf. Fig 2D in the paper).
> We will also report the results of an even stronger baseline, for which we computed the average noise level of the simulated trajectories and used this noise level during inference with the LQG. In this case, the RMSEs are still worse by roughly an order of magnitude (RMSE of 0.702 vs 0.027). Note that this information would not be readily available for real data without knowing the true parameters and therefore constitutes a strong validation, in our view.
> We will include both baseline comparisons in the supplementary material of the final version of the manuscript. We hope this sufficiently addresses the reviewer’s concerns.
>
> Thank you for catching the typos, which we will correct in the final version of the paper.

---

> ### Author Response · Authors · 2021-08-30
> **Reply to reviewer 2**
>
> *** I thank the authors for addressing my concerns with respect to the impact of the moment matching approximation and incorporating two baselines. ***
>
> We are happy this helped.
>
> *** Nonetheless, I believe that using a single experiment is not sufficient for this conference. ***
>
> We are a bit unclear about this statement by the reviewer. The present manuscript contains evaluations of the following experiments:
> 1) a single-joint reaching task with control-dependent noise [7] (details in Appendix E.1),
> 2) reaching trajectories from a previously published experiment [47],
> 3) a model of saccadic eye movements [48]
> 4) randomly-generated problems from [7] (Appendix E.3).
> Our reply also contained reference to the additional material, which we will include in the supplementary material:
> 5) experiments generating empirical distributions over trajectories estimated using Monte Carlo rollouts,
> 6) experiments for the comparison to a baseline by replacing the signal-dependent noise with additive noise.
>
> **** Due to the limited experiments and limited applicability of the method (without a simple iLQG integration, which should not be a huge effort), I believe this paper needs additional work and would like to keep the current rating as is. ***
>
> In light of the above answers, it seems to us, that the main concern is that the current manuscript does not extend the work to integrating the iLQG, which the reviewer regards as not being "a huge effort". The "forward" application of optimal control adapted to the noise characteristics of the human sensorimotor system has been one of the most widely applied methods in motor control since its publication. This, although it has not employed the iLQG. It has been very useful in modeling human and animal sensorimotor behavior, despite only allowing to guess cost functions and comparing trajectories to those observed empirically. Therefore, we think that the present paper should be extremely useful to neuroscientists and behavioral scientists and have significant impact. We don't see, how the present algorithm should be limited in applicability to these experimental data. This, to us, seems in line with the purpose of NeurIPS: "The purpose of the Neural Information Processing Systems annual meeting is to foster the exchange of research on neural information processing systems in their biological, technological, mathematical, and theoretical aspects."

---

### Author Response · Authors · 2021-08-10
**General answer to all reviewers**

We would like to thank the reviewers for their generally positive evaluation, the points they have raised and the requests they made, which have helped to improve our manuscript.

First of all, we would like to emphasize that Todorov’s model [7,8] has been one of the most widely used modeling frameworks in sensorimotor neuroscience since its publication, despite the constraints of the method (linear dynamics, quadratic costs, signal-dependent noise), which the reviewers have pointed out. The reason is that this framework has been able to explain a wide variety of experimentally observed phenomena in a single unified framework, e.g. linear movement trajectories, smooth velocity profiles, and speed-accuracy tradeoffs. It has been used also to derive computational principles that have been fundamental in explaining human and animal sensorimotor behavior and in developing novel experiments, e.g. the principle of least interaction. Its usefulness in understanding the brain’s sensorimotor control of movement in computational terms is significant.

However, the motivation for developing inverse optimal control methods for human and animal sensorimotor control based on this framework derives from the fact that currently researchers have to either use very constrained tasks for which they assume to know the cost functions employed by the subject or to flat out guess the cost function and then assume optimality of behavior. The inverse optimal control approach presented here instead allows researchers to infer the cost function thereby gaining insight into the true underlying goal of behavior, which e.g. includes the tradeoffs resulting from biomechanical and behavioral or cognitive costs. We therefore think that the current computational framework and method will be of great utility to researchers in movement science, neuroscience, and cognitive science to uncover the information processing of biological systems in terms of a computational level description.

While the inverse optimal control community might be additionally interested in methods which go beyond the limitations of the LQG setting with signal dependent noise, sensorimotor neuroscientists often have quite accurate linear models for the tasks they are studying and have spent decades in developing useful approximate linear models of limb kinematics describing body movements and even muscle dynamics. We are convinced that researchers in sensorimotor control, neuroscience, and cognitive science will benefit from having a tool at their disposal, which they can use to estimate the parameters of their optimal control models’ cost functions instead of choosing them by hand. Our extension involving the partial observability of state variables allows including latent variables in these models such as accelerations or forces, which usually are not observed directly but estimated from data as well as variables internal to the subject capturing subjective beliefs pertaining to internal cognitive state. With the present method, it is for the first time possible to carry out these inferences while taking into account the known noise characteristics of the human sensorimotor system. That this part of the model is indeed important, is now additionally quantified through two baselines, which we report at the request of one of the reviewers. We also additionally report a more thorough analysis of the impact of the moment matching approximation.

Taken together, we hope that the reviewers see the value of our contribution to the research on neural information processing systems in their biological, technological, mathematical, and theoretical aspects.

---

### Decision · Program_Chairs · 2021-09-27

**Decision:**

Accept (Poster)

**Comment:**

The paper introduces a theoretical framework for estimating cost functions in linear, quadratic, Gaussian systems with action- and state-dependent noises. Algorithms for inference are derived for the complete and partial observation cases and distinguish between the agent’s and the experimenter's inference tasks. The approach is validated on synthetic and experimental data and applied to human sequential sensorimotor behavior.

Whilst the proposed approach is a simple applications of dynamic Bayesian inference and maximum likelihood estimate, the proposed inference formulation was considered original. The theory was derived under linear, quadratic, Gaussian assumptions, but the approach is demonstrated to work well in real-world applications, such as real arm and eye movement data. The clarity of the paper could be improved, especially w.r.t. to some literature.